# LEARN BULLISH MOVES VIA EIGENCLUSTER TOKENS

## ABSTRACT

Conventional tokenization schemes in time series, such as point-wise and patch-wise methods, are poorly suited for financial time series data due to excessive token counts, sparse distributions, and heightened out-of-vocabulary risks—an issue not explicitly addressed in prior work. This paper introduces a novel tokenization approach for financial time series. By clustering scalar projections of eigenvectors from multi-window Open-High-Low-Close (OHLC) price matrices, our method generates compact and semantically meaningful tokens, enabling Transformer-based models to effectively identify next-day close price increase patterns. Extensive experiments on S&P 500 and CSI 300 datasets show our approach outperforms market baselines by 6–9% in precision, while reducing token vocabulary size to 51–101 tokens and sequence length by 75% versus point-wise.

## 1 INTRODUCTION

Current tokenization approaches in time series fall into three paradigms: point-wise, patch-wise (see Fig. 1), and variate-wise (Chen et al., 2025; Wang et al., 2024b). Point-wise methods treat each time point as a token (Zhou et al., 2021; Wu et al., 2021; Zhou et al., 2022; Liu et al., 2022), leading to redundancy and inefficiency (Dou et al., 2023). Patch-wise methods group consecutive points into segment tokens to capture local patterns (Nie et al., 2023; Zhang & Yan, 2023). Variate-wise tokenization represents an entire series as one token, emphasizing global structure but sacrificing granularity (Liu et al., 2024). Since variate-wise tokenization can be viewed as an extreme case of patch-wise (with one segment per series), we group them together in subsequent discussion. Recent advances, such as (Wang et al., 2025b; Chen et al., 2024), have extended patch-wise methods using a multi-scale approach, where time series are partitioned into patches at varying granularities. This multi-scale strategy is adopted in our work, as illustrated on the right of Fig. 1.

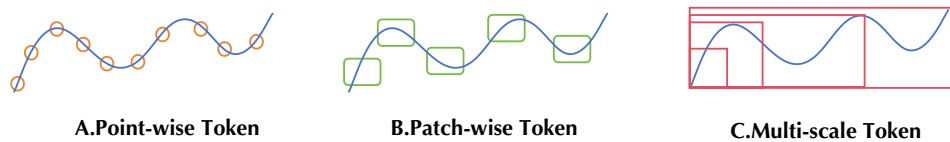

**A.Point-wise Token**    **B.Patch-wise Token**    **C.Multi-scale Token**

Figure 1: point-wise tokenization, patch-wise tokenization, multiscale discrete tokenization.

Existing tokenization methods, such as point-wise and patch-wise approaches, are often ill-suited for financial time series forecasting due to redundancy, inefficiency, and limited ability to capture meaningful temporal patterns. Inspired by recent advances in computer vision, where clustering-based tokenization has been used to extract semantically meaningful visual tokens (Liang et al., 2023; Grainger et al., 2023), we recognize the potential of domain-specific clustering strategies for financial data. Motivated by this insight and the unique characteristics of price series, we develop a novel multi-scale discrete tokenization approach, which clusters time series eigenvectors obtained from matrix transformations to generate informative tokens. This work primarily addresses three questions: *(1) Why are point-wise and patch-wise tokenizers insufficient? (2) How does our multi-scale clustering-based tokenization overcome these limitations? (3) Can the effectiveness of our method be empirically validated?* Our key contributions include:

- *Effective Pattern Recognition*: Our tokenization enables vanilla Transformers to successfully identify next-day close price increase patterns. In comprehensive evaluations across both S&P 500 and CSI 300, the identified portfolios consistently outperform market baselines by 6-9% in precision.

- *Superior Model Performance*: The proposed approach demonstrates consistent advantages over conventional token methods in predicting upward price signals across different threshold settings.
- *Computational Efficiency*: Our method significantly reduces the total number of unique tokens, with the token vocabulary size constrained within 51–101. In addition, the input token sequence length per sample is reduced by 75% compared to point-wise tokenization (from 36 to 9), enabling faster computation and inference.

The remainder of this paper proceeds as follows: In Section 2, we define the tokenization problem; while in Section 3, we review related work. Furthermore, in Sections 4 and 5, we present our methodology and architecture. We present in Section 6 the experimental results; and in Section 7 we conclude the study and the open-source code release are provided. Additional details on the experimental implementation and test results are provided in the Appendix.

## 2 PROBLEM FORMULATION

Under the embedding paradigm established in natural language processing (NLP), tokenization serves as the mechanism that maps discrete vocabulary units to continuous embedding spaces, enabling semantic structure to emerge from symbolic sequences. However, as shown in Fig. 2, applying this paradigm directly to financial time series introduces three fundamental challenges that differ markedly from NLP. For a concise mathematical perspective underlying these issues, we refer the reader to Appendix A.4.

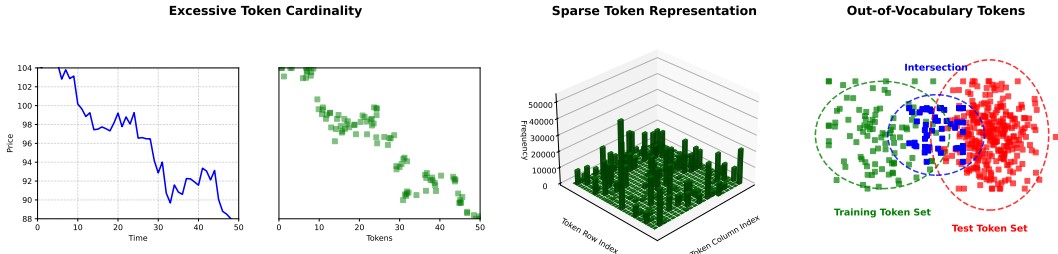

Figure 2: Visualization of financial time series tokenization challenges.

- *Excessive Token Cardinality*: Financial price series show high variability and weak periodicity, leading to an unmanageably large token space. Both point-wise and patch-wise tokenization result in exponentially growing vocabularies as decimal precision increases.
- *Sparse Token Representation*: Most tokens occur infrequently, receiving insufficient weight updates during training. This prevents learning meaningful representations in the embedding space.
- *Out-of-Vocabulary Tokens (OOV)*: Financial non-stationarity causes new tokens during testing, where $\mathcal{V}_{\text{test}} \setminus \mathcal{V}_{\text{train}} \neq \emptyset$ ($\mathcal{V}$ represents the token set). Extreme events (e.g., the negative oil price shock in 2020, unseen during training) yield tokens absent from training, limiting generalization.

Table 1: Token Counts and Out-of-Vocabulary (OOV) Analysis by Tokenization Method

| Dataset | Train/Test Period | Point-wise | Point-wise (3 dec.) | Patch-wise | Point-wise $\mathcal{V}_{\text{test}} \setminus \mathcal{V}_{\text{train}}$ | Patch-wise $\mathcal{V}_{\text{test}} \setminus \mathcal{V}_{\text{train}}$ |
|---|---|---|---|---|---|---|
| S&P500 | 00-09/11-20 | 829,240 | 607 | 230,716 | 17,784,212 | 11,531,733 |
| | 04-13/15-24 | 878,676 | 596 | 240,909 | 21,084,110 | 11,855,766 |
| CSI300 | 00-09/11-20 | 181,167 | 455 | 48,047 | 2,411,709 | 4,739,052 |
| | 04-13/15-24 | 181,334 | 454 | 48,265 | 4,188,064 | 5,797,612 |

As detailed in Table 1, we empirically validate the tokenization challenges using S&P 500 and CSI 300 data. The training set consists of OHLC data of the index itself, while the test set comprises data from the index's constituent stocks, which exhibit richer price dynamics. Each 10-day segment is normalized by day 9's closing price. We evaluate point-wise (both full precision and 3-decimal rounded) and patch-wise (encoding daily OHLC as a single token) strategies. The results demonstrate that vocabulary size grows prohibitively with decimal precision. Crucially, both methods yield unacceptably high Out-of-Vocabulary (OOV) counts due to the distributional shift

between training and testing period. This starkly contrasts with general-purpose text tokenizers like OpenAI's `cl100k-base`, which operates on a fixed vocabulary of 100,256 tokens. These critical shortcomings underscore the necessity of a robust tokenization design tailored for financial data.

## 3 RELATED WORK

The success of Transformer models in NLP has spurred interest in financial time series forecasting (Coelho e Silva et al., 2024). Current literature in finance employs two main tokenization paradigms: point-wise methods that treat the price at each time step as a token (Wang et al., 2022; Qin et al., 2017), and patch-wise approaches that group consecutive prices into segment tokens (Zeng et al., 2023; Wang et al., 2025a). As shown in Table 2, many well-known time series Transformer models fall into one of these categories. Beyond price prediction, Transformers have been adapted for multimodal financial analysis. Studies (Zhang et al., 2022; Li et al., 2022; Liu et al., 2019; Zhang et al., 2024) process textual data from news and social media to extract market sentiment, while Yang et al. (2022) combines numerical data with textual and audio information.

Table 2: Summary of Transformer-based time series models (Point-wise vs. Patch-wise)

|  | Autoformer (Wu et al.) | FEDformer (Zhou et al.) | Crossformer (Zhang & Yan) | PatchTST (Nie et al.) | iTransformer (Zhou et al.) |
|---|---|---|---|---|---|
| **Point-wise** | ✓ | ✓ |  |  |  |
| **Patch-wise** |  |  | ✓ | ✓ | ✓ |

However, the literature reveals a gap in addressing time series tokenization challenges (as detailed in Section 2). Many studies circumvent this issue by: (1) using datasets with strong periodicity where extreme values are rare (Chen et al., 2025); (2) employing small test sets to avoid extreme scenarios (Xu et al., 2025); (3) using z-score normalization that reduces value dispersion (Zhu et al., 2025); 4) replacing the embedding layer with either a simple linear projection or a convolutional mapping, thereby bypassing the tokenization problem entirely (Nie et al., 2023; Wu et al., 2021).

Our paper presents a data-centric critique demonstrating how current tokenization approaches are fundamentally mismatched to the unique characteristics of financial time series. To address these critical limitations, we introduce a spectral clustering strategy (Xiang & Gong, 2008; Tai et al., 2022) that first constructs a multi-scale representation from the dataset, then extracts its eigenvectors and performs clustering to guide a more adaptive, fine-grained token segmentation.

## 4 METHODOLOGY

This section presents our eigen-cluster tokenization approach, comprising four key components: (1) prefix-window matrix representation, (2) matrix transformation and eigenvector computation, (3) scalar projection and cluster-based tokenization, and (4) identification and interpretation of the most bullish cluster. The overall workflow, which integrates multi-scale patching, eigendecomposition, and clustering, is illustrated in Fig. 3 using an example with $n = 10$.

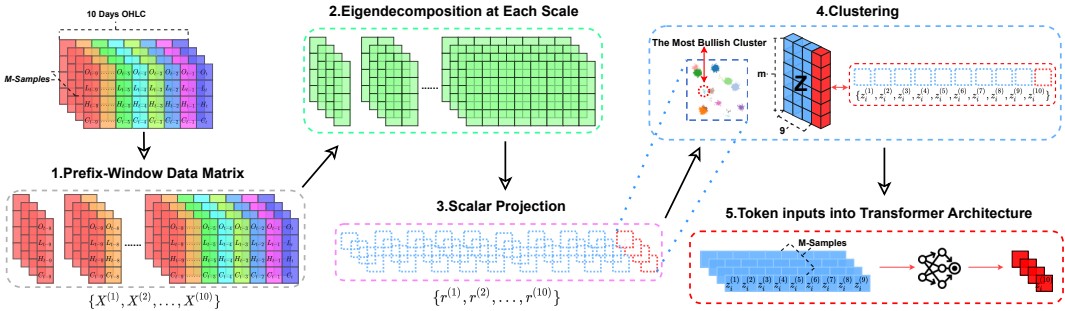

Figure 3: Workflow of multi-scale tokenization with eigendecomposition and clustering.

## 4.1 PREFIX-WINDOW DATA MATRIX REPRESENTATION

To extract multi-scale temporal features, we construct a sequence of data matrices based on *prefix windows*. Unlike sliding windows, here the windows are nested and grow forward from the starting point $t - n + 1$ until $t$. Specifically, the $k$-th window ($k = 1, 2, \ldots, n$) covers the time range from $t - n + 1$ to $t - n + k$. Here, $k$ can be as small as 1, corresponding to a single-day window, or as large as $n$, corresponding to the full prefix ending at the current time $t$. Hence, only the largest window ($k = n$) includes the current time $t$. For each window length $k$, we define a data matrix $X^{(k)} \in \mathbb{R}^{m \times 4k}$, where each row corresponds to one sample and is formed by concatenating the OHLC vectors of the $k$ consecutive days in that prefix window. Formally,

$$\boldsymbol{X}^{(k)} = \begin{bmatrix} \mathbf{x}_1^{(t-n+1)} & \mathbf{x}_1^{(t-n+2)} & \cdots & \mathbf{x}_1^{(t-n+k)} \\ \mathbf{x}_2^{(t-n+1)} & \mathbf{x}_2^{(t-n+2)} & \cdots & \mathbf{x}_2^{(t-n+k)} \\ \vdots & \vdots & \ddots & \vdots \\ \mathbf{x}_m^{(t-n+1)} & \mathbf{x}_m^{(t-n+2)} & \cdots & \mathbf{x}_m^{(t-n+k)} \end{bmatrix}. \tag{1}$$

Each vector in the matrix is defined as

$$\mathbf{x}_i^{(t-n+j)} = \begin{bmatrix} \dfrac{O_i^{(t-n+j)}}{C_i^{(t-n+k-1)}} & \dfrac{H_i^{(t-n+j)}}{C_i^{(t-n+k-1)}} & \dfrac{L_i^{(t-n+j)}}{C_i^{(t-n+k-1)}} & \dfrac{C_i^{(t-n+j)}}{C_i^{(t-n+k-1)}} \end{bmatrix}, j = 1, 2, \ldots, k. \tag{2}$$

Each row is normalized by the closing price at time $t - n + k - 1$, which serves as the denominator for all values in that sample. This construction produces a family of matrices $\{X^{(1)}, X^{(2)}, \ldots, X^{(n)}\}$, where only $X^{(n)}$ contains the most recent observation $t$, while smaller matrices correspond to shorter historical prefixes. This multiscale structure is inspired by prior work on hierarchical time series modeling, such as TimeMixer (Wang et al., 2024a; 2025b), which demonstrates the effectiveness of multi-resolution temporal decomposition in capturing complex temporal dependencies.

## 4.2 EIGENDECOMPOSITION AT EACH SCALE

To extract dominant patterns from the feature matrices at each scale, we perform eigendecomposition on the covariance matrix of the centered data matrix corresponding to a specific prefix window. As an example, we consider the largest window $k = n$, which covers the full prefix ending at time $t$.

Let $\boldsymbol{X}^{(n)} \in \mathbb{R}^{m \times 4n}$ be the data matrix for the largest prefix. We first compute the mean vector and the centered matrix:

$$\boldsymbol{\mu}^{(n)} = \frac{1}{m} \sum_{i=1}^{m} \boldsymbol{x}_i^{(n)} \in \mathbb{R}^{4n}, \quad \boldsymbol{X}_{\text{centered}}^{(n)} = \boldsymbol{X}^{(n)} - \mathbf{1}(\boldsymbol{\mu}^{(n)})^\top \in \mathbb{R}^{m \times 4n}, \tag{3}$$

where $\mathbf{1}$ is an $m$-dimensional column vector of ones and $\boldsymbol{x}_i^{(n)}$ denotes the $i$-th row of $X^{(n)}$. The empirical covariance matrix is then

$$\boldsymbol{A}^{(n)} = \frac{1}{m} (\boldsymbol{X}_{\text{centered}}^{(n)})^\top \boldsymbol{X}_{\text{centered}}^{(n)} \in \mathbb{R}^{4n \times 4n}. \tag{4}$$

Solving the eigenvalue problem $\boldsymbol{A}^{(n)} \boldsymbol{v}_i^{(n)} = \lambda_i^{(n)} \boldsymbol{v}_i^{(n)}$ yields eigenpairs $(\lambda_i^{(n)}, \boldsymbol{v}_i^{(n)})$, where $\lambda_i^{(n)}$ represents the variance explained by the $i$-th principal direction $\boldsymbol{v}_i^{(n)}$. We use all eigenvectors to form the projection matrix:

$$\boldsymbol{W}^{(n)} = [\boldsymbol{v}_1^{(n)} \, \boldsymbol{v}_2^{(n)} \, \cdots \, \boldsymbol{v}_{4n}^{(n)}] \in \mathbb{R}^{4n \times 4n}, \quad \widetilde{\boldsymbol{X}}^{(n)} = \boldsymbol{X}_{\text{centered}}^{(n)} \boldsymbol{W}^{(n)} \in \mathbb{R}^{m \times 4n}. \tag{5}$$

This procedure can be applied to any prefix window $k = 1, 2, \ldots, n$ to obtain scale-specific principal components, enabling multi-scale temporal pattern extraction. A similar operation has also been adopted in Tai et al. (2022). Notably, the eigendecomposition step is effectively a Principal Component Analysis (PCA) operation; however, unlike conventional PCA, we retain all eigenvectors rather than selecting only the leading components. Prior work has shown that PCA can enhance K-means clustering by maximizing variance and making cluster directions more separable (Zha et al., 2001; Ding & He, 2004).

### 4.3 SCALAR PROJECTION AND CLUSTERING

After obtaining the eigendecomposed and projected feature matrices $\widetilde{\boldsymbol{X}}^{(k)} \in \mathbb{R}^{m \times 4k}$ at each scale $k = 0, 1, \ldots, n$, we apply a monotonic transformation to compress each row into a single interpretable scalar. This mapping, which we refer to as a scalar projection, preserves both directional and magnitude characteristics of the original high-dimensional representation. The use of the trigonometric function $\sin(\cdot)$ provides a bounded directional signal in $[-1, 1]$, while $\|\cdot\|_2$ encodes its corresponding amplitude. This design enables subsequent visualization and supports conversion from multivariate representations to a one-dimensional interpretable form. Specifically, for the $i$-th sample at scale $k$, we define:

$$r_i^{(k)} = \left( \frac{1}{4k} \sum_{j=1}^{4k} \sin(\widetilde{x}_{ij}^{(k)}) \right) \cdot \left\| \widetilde{\boldsymbol{x}}_i^{(k)} \right\|_2 , \tag{6}$$

where $\widetilde{\boldsymbol{x}}_i^{(k)}$ denotes the $i$-th row of $\widetilde{\boldsymbol{X}}^{(k)}$ for $i = 1, \ldots, m$. The $j$-th entry of $\widetilde{\boldsymbol{x}}_i^{(k)}$ is written as $\widetilde{x}_{ij}^{(k)}$, corresponding to the value in the $i$-th row and $j$-th column of $\widetilde{\boldsymbol{X}}^{(k)}$. This yields a scalar vector $\mathbf{r}^{(k)} = [r_1^{(k)}, r_2^{(k)}, \ldots, r_m^{(k)}]^\top \in \mathbb{R}^m$ for clustering. We then apply one-dimensional $K$-means clustering to $\mathbf{r}^{(k)}$, producing clusters $\{\mathcal{S}_1^{(k)}, \ldots, \mathcal{S}_K^{(k)}\}$ by minimizing the within-cluster variance:

$$\min_{\{\mathcal{S}_1^{(k)}, \ldots, \mathcal{S}_K^{(k)}\}} \sum_{j=1}^{K} \sum_{r_i^{(k)} \in \mathcal{S}_j^{(k)}} \left( r_i^{(k)} - \mu_j^{(k)} \right)^2 , \tag{7}$$

where $\mu_j^{(k)}$ is the centroid of cluster $\mathcal{S}_j^{(k)}$. Each sample is assigned a cluster token $z_i^{(k)} \in \{1, 2, \ldots, K\}$ based on the closest centroid. Repeating this procedure for all scales $k = 0, 1, \ldots, n$ yields a multi-resolution token vector for each sample, and stacking all per-sample vectors forms the final token matrix (see Fig. 3 ):

$$\mathbf{z}_i = \begin{bmatrix} z_i^{(1)} & z_i^{(2)} & \cdots & z_i^{(n)} \end{bmatrix} \in \mathbb{R}^n, \quad \mathbf{Z} = \begin{bmatrix} \mathbf{z}_1 & \mathbf{z}_2 & \cdots & \mathbf{z}_m \end{bmatrix} \in \mathbb{R}^{m \times n}. \tag{8}$$

This multi-scale tokenized representation captures dominant temporal patterns in price dynamics, making it directly compatible with Transformer architectures. By clustering sequences into $K$ tokens, this approach not only mitigates OOV issues but also compresses the vocabulary. This consolidation significantly increases the frequency of each token's occurrence, thereby directly alleviating the sparse token problem alongside the excessive token problem.

### 4.4 THE BULLISH CLUSTER

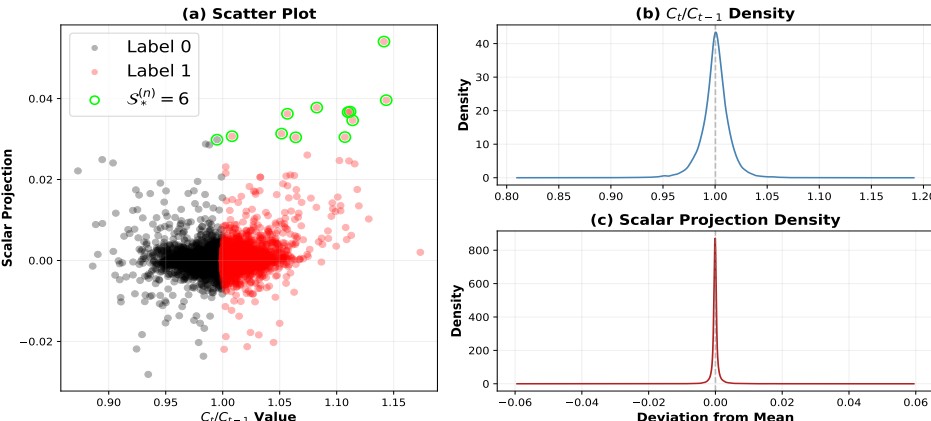

Figure 4: Three-panel visualization: (a) scalar projection vs. $C_t/C_{t-1}$ scatter with the bullish cluster highlight; (b) density of $C_t/C_{t-1}$; (c) density of scalar projections $r_i^{(n)}$.

After obtaining the multi-scale token representation, the prediction task becomes estimating the $n$-th token $\hat{z}_i^{(n)} \in \{1, 2, \ldots, K\}$ from the preceding $n-1$ tokens $[z_i^{(1)}, z_i^{(2)}, \ldots, z_i^{(n-1)}]$. Each predicted token $z_i^{(n)}$ maps to one of the $K$ clusters derived from projected price trajectories. We define $\mathcal{S}_*^{(n)}$ as the most bullish cluster according to the following criterion:

$$\mathcal{S}_*^{(n)} := \underset{\mathcal{S} \in \{\mathcal{S}_1^{(n)}, \ldots, \mathcal{S}_K^{(n)}\}}{\arg \min} |\mathcal{S}| \quad \text{s.t.} \quad \left| \left\{ i \in \mathcal{S} : C_t^{(i)} > C_{t-1}^{(i)} \right\} \right| > \left| \left\{ i \in \mathcal{S} : C_t^{(i)} < C_{t-1}^{(i)} \right\} \right| \quad (9)$$

We select the smallest cluster with the highest concentration of rising patterns, ensuring $\mathcal{S}_*^{(n)}$ contains the most reliable bullish signals. As confirmed in Fig. 4(a), this cluster shows nearly perfect separation, with almost all samples satisfying $C_t > C_{t-1}$. Subfigures (b) and (c) show kernel densities of normalized $C_t/C_{t-1}$ and scalar projections, respectively. In addition, we apply Synthetic Minority Over-sampling Technique (SMOTE) (Chawla et al., 2002) to mitigate class imbalance (see Appendix A.5).

## 5 TRANSFORMER ARCHITECTURE AND SETTING

This section details our multi-scale tokenization framework, the method for selecting optimal K-means clusters, and the vanilla Transformer architecture employed.

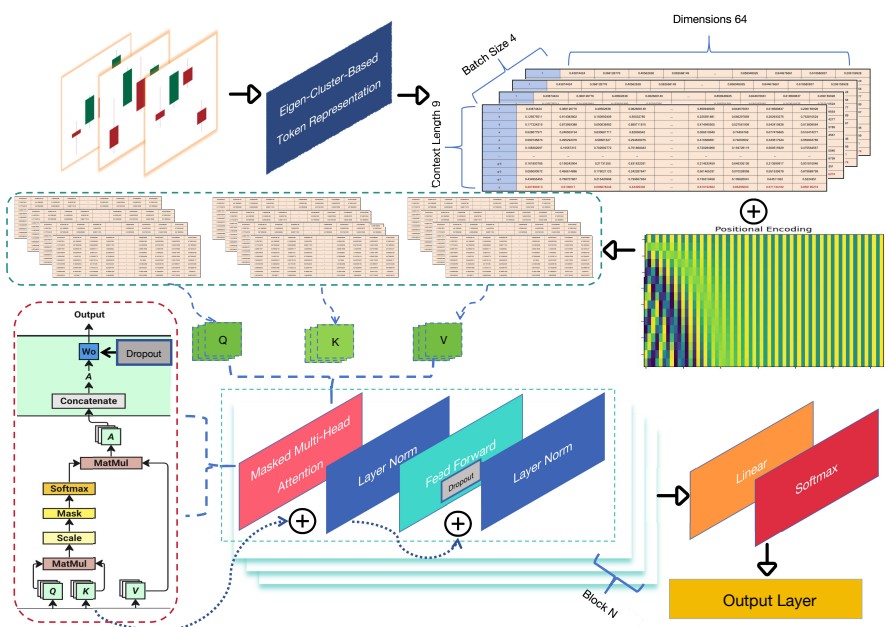

Figure 5: Workflow of Transformer Architecture.

### 5.1 MULTI-SCALE SETTING AND OPTIMAL K CLUSTERS

We construct a 10-day OHLC matrix $\boldsymbol{X} \in \mathbb{R}^{m \times 40}$. Nested prefix matrices $\boldsymbol{X}^{(k)} \in \mathbb{R}^{m \times 4k}$ for $k = 1, \ldots, 10$ contain the first $k$ days, with $\boldsymbol{X}^{(10)}$ including all days to target $t$. Each $\boldsymbol{X}^{(k)}$ has response vector $\mathbf{r}^{(k)} \in \mathbb{R}^m$. For $\mathbf{r}^{(1)}$ to $\mathbf{r}^{(9)}$, we use fixed $K = 5$ or $10$, yielding two variants: Ours-5 and Ours-10. For $\mathbf{r}^{(10)}$, the optimal $K^*$ is determined by maximizing the score over $K \in [5, 20]$.

$$\underset{K \in \{5, \ldots, 20\}}{\arg \max} \left[ \text{Score} = 25.0 \cdot p_*^{(10)} - 0.8 \cdot \left( \frac{50}{n_*^{(10)}} \right) - 0.2 \cdot K \right] \quad (10)$$

where $p_*^{(10)} \equiv \Pr(C_t/C_{t-1} > 1 | \mathcal{S}_*^{(10)})$: bullish probability in cluster; $n_*^{(10)} \equiv |\mathcal{S}_*^{(10)}|$: cluster size; $K$: number of clusters. The weights reflect our prioritization (as detailed in Appendix A.6): (1) Strong emphasis on bullish rates (25.0) to promote upward-oriented samples; (2) Moderate concern

for larger cluster size (0.8) to avoid small, unreliable clusters; (3) Mild preference for smaller $K$ (0.2) to reduce the overall token count. Table 3 shows the optimal $K$ values for $\mathbf{r}^{(10)}$ and the most bullish cluster $\mathcal{S}_*^{(10)}$ that we adopted for different markets and training periods.

Table 3: Optimal $K$ for $\mathbf{r}^{(10)}$ and Bullish Cluster $\mathcal{S}_*^{(10)}$

| S&P500 | | | | CSI300 | | | |
|--------|-----------------|-----|-------------------|--------|-----------------|-----|-------------------|
| Market | Training Period | $K$ | $\mathcal{S}_*^{(10)}$ | Market | Training Period | $K$ | $\mathcal{S}_*^{(10)}$ |
| S&P500 | 2000–2009 | 6 | 4 | CSI300 | 2000–2009 | 7 | 6 |
| S&P500 | 2004–2013 | 11 | 6 | CSI300 | 2004–2013 | 9 | 2 |

## 5.2 TRANSFORMER ARCHITECTURE

In this study, we employ a decoder-only Transformer as a shared architecture for all tokenization methods under a controlled setup, ensuring that performance differences arise from the token representation rather than architectural variance. As shown in Fig. 5, the input time series undergoes normalization, multiscale segmentation, eigendecomposition, and clustering, producing a sequence of 9 discrete cluster tokens $[z_i^{(1)}, z_i^{(2)}, \ldots, z_i^{(9)}]$ used to predict the next token $z_i^{(10)}$. Each token is embedded into a 64-dimensional vector ($d_{\text{model}} = 64$), forming a $9 \times 64$ input matrix, with sinusoidal positional encodings (Vaswani et al., 2017) added to preserve temporal order. Notably, our embedding layer follows the conventional NLP-style discrete lookup paradigm. This contrasts with linear projection embeddings (e.g., PatchTST) or convolutional embeddings (e.g., Autoformer), where inputs are mapped into continuous representations. In those architectures, the notion of symbolic "tokens" no longer exists, and the model no longer processes relationships between discrete tokens.

The embedding matrix is processed by an 8-layer decoder-only Transformer with 4 attention heads per layer. The architecture follows the standard design with residual connections and layer normalization before both attention and feed-forward modules. Dropout (rate 0.1) is applied within these modules to reduce overfitting. The output is projected via a linear layer followed by softmax to produce a distribution over $K$ cluster tokens. Hyperparameters were selected considering dataset size and computational resources: batch size 4, $d_{\text{model}} = 64$, 8 decoder blocks, 4 attention heads, learning rate $1 \times 10^{-3}$, dropout 0.1, and 2000 training epochs. The model used the AdamW optimizer, cross-entropy loss, and random seed 1337.

## 6 EMPIRICAL RESULTS

This section presents our experimental data and evaluation of different approaches for stock upward recognition. Our dataset consists of two distinct groups for cross-market evaluation:

Table 4: Data Split Time Periods with Token Vocabulary Sizes

| Train | Token Vocabulary Size | | Validation | Test | Stocks |
|---------|---------------|----------------|------------|---------|------------------|
| | Ours-5 | Ours-10 | | | |
| 2000–09 | 51(US), 52(CN) | 96(US), 97(CN) | 2010 | 2011–20 | 493(US), 288(CN) |
| 2004–13 | 56(US), 54(CN) | 101(US), 99(CN) | 2014 | 2015–24 | 501(US), 300(CN) |

*U.S. and Chinese (CN) Markets:* We evaluate our approach on two distinct markets with different training sets. For the U.S. market, we train on 10 major global indices and test on S&P 500 constituent stocks. For the Chinese market, we train on two domestic indices and test on CSI 300 constituent stocks[1]. This design offers two advantages: (1) indices better capture overall market trends than individual stocks; (2) using completely different datasets for training and testing rigorously evaluates model robustness. The chronological splitting follows two complete market cycles as detailed in Table 4. Validation are used exclusively for early stopping to prevent overfitting.

---

[1]U.S. data sources: S&P 500, NASDAQ, Hang Seng, Dow Jones, CAC 40, DAXI, Nikkei 225, KOSPI, BSE, EURO STOXX 50 (from https://pypi.org/project/yfinance/). Chinese data sources: SSEC (Shanghai Composite) and SZSC (Shenzhen Component) (from Tushare: https://tushare.pro/).

*Evaluation Metrics:* We evaluate model performance using Precision = TP/(TP + FP). Here, TP (True Positives) are correct predictions of upward movement ($C_t/C_{t-1} > 1$), and FP (False Positives) are incorrect upward predictions. This metric interprets the proportion of correct upward predictions. Unlike regression metrics (MSE, RMSE) that quantify magnitude errors, precision captures directional accuracy—essential for trading where even minor misdirection results in losses.

## 6.1 COMPARISON OF TOKENIZATION METHODS

As summarized in Table 5, we evaluate three tokenization methods within a unified framework. Point-wise and patch-wise methods use the same normalization procedure (division by $C_{t-1}$) rather than conventional z-score normalization, as the global statistics (e.g., max/min) of the entire trading data are unavailable in practice. Each method predicts the next token, which is mapped to a numerical value for thresholding at $\tau \in [1.00, 1.03]$. Our method uses the normalization in Equation 2 and predicts token cluster membership. All tokenization methods employ the same vanilla Transformer architecture (Refer to Section 5.2), allowing direct observation of how different tokenization strategies impact the Transformer's performance.

OOV tokens are processed using: (1) KDTree nearest-neighbor retrieval from `scikit-learn` for patch-wise tokenization, and (2) exact Euclidean search within a reduced vocabulary of 400–600 tokens for point-wise tokenization. The KDTree approach is adopted for patch-wise tokenization to ensure computational efficiency, as performing exact Euclidean matching over a large token set (e.g., 240,000+) would be prohibitively slow. Meanwhile, the point-wise method operates on a much smaller token set, making exact Euclidean search feasible. The 3-decimal rounding applied in point-wise tokenization serves two reasons: it significantly shrinks the token vocabulary to avoid excessive memory consumption, and it alleviates extreme token frequency imbalances that would otherwise bias predictions toward frequent values (such as 1).

Table 5: Comparison of Tokenization Strategies

| Method | Input | Output | Configuration |
|--------|-------|--------|---------------|
| Ours | 9 tokens | $\mathbb{I}(\hat{z}_i^{(10)} \in \mathcal{S}_*^{(10)})$ | Employs 9 tokens to predict the $\hat{z}_i^{(10)}$, we evaluate two configurations: `Ours-5&10`. |
| Point-wise | 36 tokens | $\mathbb{I}(\hat{z}_i^{(37)} > \tau)$ | 3-decimal rounding reduces tokens to 454–607; each time point's OHLC prices as separate tokens, using 36 tokens (9 days × 4). |
| Patch-wise | 9 tokens | $\mathbb{I}(\hat{z}_i^{(10)}[\text{close}] > \tau)$ | Treats each trading day's OHLC as a single token segment, using 9 tokens to predict the close within $\hat{z}_i^{(10)}$. |

## 6.2 PERFORMANCE ACROSS TOKEN METHODS

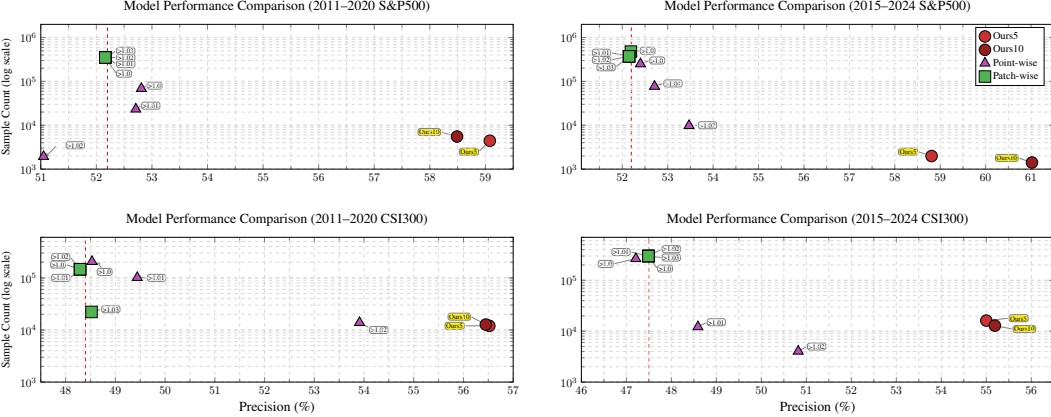

Figure 6: Model performance comparison on S&P500 and CSI300 constituent stocks. The red dashed line: market baselines (TP rates in test). Models with sample counts below 1000 are omitted.

Fig. 6 demonstrates the clear superiority of our eigen-cluster tokenization, outperforming both point-wise and patch-wise methods by 2.6–8.8% precision across all thresholds and evaluation periods. This improvement arises from resolving fundamental limitations in conventional designs: excessive token counts (point-wise: 400–600, patch-wise: 240,000+) and the dependence on unoptimized out-of-vocabulary substitutions that introduce approximation errors. The full numerical results corresponding to this figure are provided in Appendix A.9.

Our models consistently achieve 6 to 9% higher precision than market baselines (the red dash lines), which represent the proportion of truly bullish samples in the test set and thus correspond to a random-guessing strategy. In contrast, conventional methods struggle to exceed these baselines when $\tau > 1.0$ and fail to maintain high precision at $\tau > 1.03$. Notably, patch-wise models show little change across thresholds, as the vast token space dilutes probability distributions and reduces sensitivity. These results highlight tokenization, rather than model complexity, as the primary bottleneck in financial time series forecasting with Transformers. For a practical evaluation, Appendix A.8 presents a trading backtest using the `Ours5` model over the period 2015–2024.

To evaluate computational efficiency, we measured inference times for predicting 1,202,584 samples from the S&P 500 test set (2015–2024) on Google Colab platform with T4 GPU (15,360 MiB VRAM), as detailed in Table 6. Our method achieves $27\times$ and $16\times$ speedups over patch-wise and point-wise approaches respectively, resulting from dramatically reduced token vocabulary (51–101 vs. 400–600 vs. 240,000+) and shorter input sequences (9 vs. 36 tokens). In addition to the Transformer architecture evaluation, we conducted comprehensive experiments comparing with traditional machine learning models; detailed results are provided in Appendix A.7.

Table 6: Inference Time Comparison on S&P 500 Test Set (2015–2024)

| Method | Time | Speedup | Method | Time | Speedup |
|---|---|---|---|---|---|
| Patch-wise | 4:55 (295 s) | 1.0× | Ours-5 | 0:11 (11 s) | 26.8× |
| Point-wise | 2:58 (178 s) | 1.7× | Ours-10 | 0:11 (11 s) | 26.8× |

## 6.3 THE IMPACT OF TOKEN LENGTH AND ARCHITECTURAL DEPTH

We also conduct an experiment to examine the effect of input token length. Specifically, we test four configurations using 5, 10, 15, and 30 tokens as input (Table 7). The experiments are carried out on the S&P 500 dataset covering 2015–2024. The results show that increasing the input length does not improve performance; instead, shorter historical windows (e.g., 4 input tokens predicting the next one) yield the best precision, suggesting that local market patterns are more informative for short-term forecasting.

Table 7: Precision performance across different token counts.

| Total Tokens | Input Tokens | Predicted Tokens | Precision (%) |
|---|---|---|---|
| 5 | 4 | 1 | **61.99** |
| 10 | 9 | 1 | 61.02 |
| 15 | 14 | 1 | 54.94 |
| 30 | 29 | 1 | 52.92 |

To further examine architectural sensitivity, we compare our 8-layer Transformer against a 2-layer version and a 1-layer LSTM baseline on the same S&P 500 test set. As shown by the precision results below, deeper Transformer architectures yield moderate improvements over shallower variants and substantially outperform the LSTM baseline.

Table 8: Precision comparison of model architectures.

| Model | Precision (%) |
|---|---|
| 8-layer Transformer (Ours-10) | **61.02** |
| 2-layer Transformer (Ours-10) | 59.63 |
| 1-layer LSTM | 54.86 |

## 6.4 CORRELATION AMONG THE TEST STOCKS

To examine whether our method performs differently across stocks with different levels of representativeness in the index, we conducted an additional experiment using the S&P 500 constituents from

2011–2020 as the test set. We grouped the constituents based on their index weights and evaluated the prediction accuracy for four segments: the top 10%, top 30%, bottom 30%, and bottom 10% by index weight. Table 9 reports the results. We observe that the method achieves higher precision for stocks with larger benchmark weights, whereas the precision decreases for the smaller-weight stocks. This suggests that our token-based up-movement prediction method is more effective for stocks that are more strongly related to the overall market index.

Table 9: Prediction Precision Across S&P 500 Constituents (Grouped by Index Weight)

| Constituent Group | Precision (%) |
|---|---|
| Top 10% by weight | 60.29 |
| Top 30% by weight | 61.35 |
| Bottom 30% by weight | 58.28 |
| Bottom 10% by weight | 56.81 |

## 6.5 Ablation Experiment

We perform ablation studies on the 2011–2020 dataset to evaluate three main components of our tokenization framework: (1) single-scale patch or multi-scale representation, (2) eigenvector projection, and (3) cluster-based tokenization. As shown in Table 10, removing any of these components leads to a clear decrease in precision on both the S&P500 and CSI300 datasets, demonstrating their complementary contributions.

Table 10: Ablation Results (Precision %, 2011–2020)

| Patch | Multi-scale | Eigenvectors | Clustering | S&P500 | | CSI300 | |
|---|---|---|---|---|---|---|---|
| | | | | Ours5 | Ours10 | Ours5 | Ours10 |
| ✗ | ✓ | ✓ | ✓ | **59.08** | **58.49** | **56.52** | **56.45** |
| ✗ | ✓ | ✓ | ✗ | 52.20 | 52.09 | 48.40 | 48.43 |
| ✗ | ✓ | ✗ | ✓ | 54.58 | 54.57 | 55.72 | 56.13 |
| ✓ | ✗ | ✓ | ✓ | 52.80 | | 48.62 | |

Note: ✗ indicates the component is ablated, ✓ indicates the component is included.

Clustering is critical—without it, tokenization degenerates into patch-wise schemes with excessive tokens and out-of-vocabulary issues, causing substantial precision drops. Eigendecomposition further enhances the expressiveness of bullish clusters; removing it yields moderate precision reductions, particularly on the S&P500. Moreover, replacing multi-scale patching with single-scale day-level patch tokens (i.e., using one day's OHLC as a segment token) markedly degrades precision on both S&P500 and CSI300. Notably, the choice of K for the first nine prefix windows (5 or 10) has minimal impact on performance, with only small differences between `Ours5` and `Ours10`.

## 7 Conclusion

This paper addresses key challenges in applying Transformers to stock forecasting, noting that while an increasing number of studies leverage Transformers for time series prediction, the fundamental issue of tokenization remains largely unresolved. We identify core limitations in conventional point-wise and patch-wise tokenization approaches: *Excessive Token Vocabulary*, *Sparse Token Distributions*, and *Out-of-Vocabulary Token Issues*. To overcome these challenges, we propose a novel multiscale cluster discrete tokenization framework that: (1) employs multiscale segmentation of time series to capture hierarchical patterns, (2) utilizes eigenvector projection for effective feature extraction, and (3) implements clustering-based tokenization to achieve semantic-aware discretization. Experimental results demonstrate both the shortcomings of conventional tokenization methods and the effectiveness of our approach. Our method enables Transformers to match or outperform classical baselines across various prediction thresholds and market conditions. Ablation studies further confirm the critical contribution of each component to the overall performance. Overall, we emphasize that for financial time series applications, tokenization should prioritize reducing token counts while preserving semantic meaning. The implementation code for this study is available at: https://github.com/MasterBeard/EigenCluster-Tokenization-for-Financial-Transformers.

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

## A APPENDIX

This appendix provides additional discussions on several technical components that were only briefly referenced in the main text, and includes a disclosure regarding the use of large language models (LLMs) in the preparation of this manuscript.

First, Appendix A.1 examines whether the proposed tokenization design introduces risks of overfitting under the learning framework by training and validation loss curves. Appendix A.2 extends the discussion on clustering by comparing alternative methods, and further justifies the selection of $k$-means based on its ability to obtain the most informative and actionable bullish cluster. Appendix A.3 explains why tokenization approaches widely adopted in vision—such as VQ-VAE and discrete autoencoders—are not applicable to our financial time-series setting.

In addition to these points, we provide a mathematical discussion of the three fundamental tokenization challenges introduced in Section 2. Second, we elaborate on the role of SMOTE in addressing class imbalance within our predictive framework. Third, we detail the scoring function in Eq. 10 and explain the rationale behind the weighting scheme used to determine the optimal number of clusters $K^*$. Fourth, we present a comparative analysis demonstrating that our vanilla Transformer models with eigencluster-based tokens generally achieve higher precision than traditional baselines, while also acknowledging several extreme cases where conventional models outperform our predictions. Finally, we report a historical backtest of the `Ours5` model over the 2015–2024 period, as reported in Appendix A.8, to evaluate the practical utility of the proposed predictive framework.

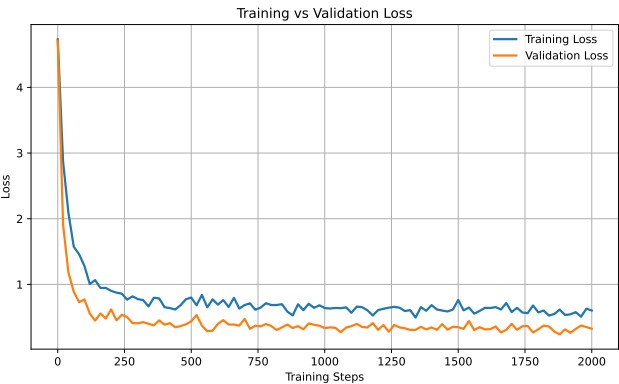

Figure 7: Training and validation loss curves for the proposed model (S&P500 2004–2013).

## A.1 Loss Curves

We have included the training and validation loss curves in the following (see Fig. 7). As illustrated in the figure, both training and validation losses consistently decrease throughout the entire training process, and no late-stage increase in validation loss is observed. This indicates that the model is learning generalizable patterns rather than memorizing the training data.

## A.2 K-means vs GMM

We tested various automatic clustering approaches, including Gaussian Mixture Models (GMM) with AIC/BIC selection and Bayesian Gaussian Mixture Models (Bayesian GMM). On the S&P 500 training set (2004–2013), the bullish cluster from these methods exhibited only moderate bullish probabilities (75%, 68%, and 61%, respectively). In contrast, using 1D K-means with $k = 11$ produced a top cluster with a bullish probability of 92%, which is substantially stronger. Therefore, we chose 1D K-means as our main clustering method to obtain the most informative and actionable bullish cluster.

Table 11: Comparison of different clustering methods on the SP500 training set (2004-2013).

| Clustering Method | Selection | Bullish Cluster (%) |
|---|---|---|
| GMM | AIC | 75% |
| GMM | BIC | 68% |
| Bayesian GMM | Default | 61% |
| K-means | k=11 | 92% |

## A.3 Why VQ-VAE and Discrete Autoencoders Collapse on Financial Data

We experimented with VQ-VAE and discrete autoencoders as alternative tokenization approaches; however, these methods consistently collapse when applied to normalized financial price features. Due to the extremely low variance of normalized inputs, almost all samples lie near a single mode, causing the quantization stage to map the entire dataset to one identical discrete token and preventing meaningful token diversity.

Our input features are normalized financial series with very limited dispersion:

$$x_{i,j} \in \mathbb{R}, \qquad \mu \approx 1.0005, \;\; \sigma \approx 0.0328, \tag{11}$$

where $x_{i,j}$ denotes the $j$-th feature of the $i$-th sample, $\mu$ is the empirical mean, and $\sigma$ is the empirical standard deviation. Thus,

$$x_{i,j} = \mu + \varepsilon_{i,j}, \qquad |\varepsilon_{i,j}| \ll 1, \tag{12}$$

meaning the deviations $\varepsilon_{i,j}$ are extremely small. For any encoder $f_\theta$ in VQ-VAE or a discrete autoencoder, such nearly constant inputs produce almost identical latent vectors:

$$z_i = f_\theta(x_i) \approx z_j, \quad \forall i, j, \tag{13}$$

where $z_i$ is the latent representation of sample $i$. Consequently, the quantization layer (e.g., Gumbel-Softmax or VQ codebook) receives almost the same logits for every sample. For a linear projection of the latent vector into code logits, we have

$$\ell_i = W z_i + b \approx \ell_j, \tag{14}$$

where $z_i \in \mathbb{R}^d$ is the latent representation of sample $i$, $W \in \mathbb{R}^{K \times d}$ and $b \in \mathbb{R}^K$ are the learnable weight matrix and bias projecting $z_i$ to $K$ logits, and $\ell_i \in \mathbb{R}^K$ is the vector of logits corresponding to the $K$ discrete codebook entries. Since all latent vectors are nearly identical ($z_i \approx z_j$ for all $i, j$), the logits are also almost the same ($\ell_i \approx \ell_j$). As a result, the quantization step collapses all samples to the same code index:

$$\arg\max_k \ell_{i,k} = k_0, \quad \text{for all } i, \tag{15}$$

where $\ell_{i,k}$ denotes the $k$-th component of $\ell_i$, and $k_0 \in \{1, \ldots, K\}$ is the single code index assigned to all samples. Therefore, the degeneration is not a failure of VQ-VAE or discrete autoencoders, but

a direct consequence of the low-variance, unimodal structure of normalized financial data, which lacks the multimodal geometry needed for meaningful discrete token learning.

For transparency and reproducibility, we provide a Colab notebook in an anonymous GitHub repository containing our VQ-VAE and discrete autoencoder experiments. The notebook can be executed directly to verify the observed behavior: https://github.com/MasterBeard/EigenCluster-Tokenization-for-Financial-Transformers/blob/main/ICLR_review3_VQ_VAE.ipynb

### A.4 Challenges in Financial Tokenization

*Excessive Token Cardinality:* Financial price series exhibit high variability and weak periodicity, which leads to a token space that grows approximately as an exponential function of the decimal precision. Formally, the vocabulary size grows as

$$|\mathcal{V}| \propto B^d,\tag{16}$$

where $B$ is the base of representation (e.g., $B = 10$ for decimal prices) and $d$ is the decimal precision. This relationship demonstrates that the number of unique tokens scales exponentially with precision, making the tokenization approach computationally intractable at high precision levels.

*Update Sparsity:* The fundamental issue is that each token's embedding vector is only updated when that specific token appears in the training data. Therefore, a token $v$ that occurs only $n$ times receives exactly $n$ updates to its embedding $\mathbf{e}_v$. For a rare token (e.g., $n = 1$), this results in a single, ineffective gradient step:

$$\mathbf{e}_v^{(final)} \approx \mathbf{e}_v^{(initial)} - \eta \nabla \mathbf{e}_v \mathcal{L}.\tag{17}$$

where $\eta$ is the learning rate and $\nabla \mathbf{e}_v \mathcal{L}$ is the gradient from the single occurrence of token $v$. This single update is negligible compared to the thousands of updates received by frequent tokens. Consequently, the embeddings of rare tokens remain poorly optimized and fail to learn meaningful representations.

*Out-of-Vocabulary Tokens (OOV):* Due to financial non-stationarity, novel patterns appear at test time that were never observed during training. This yields tokens outside the training vocabulary:

$$\mathcal{V}_{\text{test}} \setminus \mathcal{V}_{\text{train}} \neq \emptyset,\tag{18}$$

where $\mathcal{V}_{\text{train}}$ and $\mathcal{V}_{\text{test}}$ denote the training and testing vocabularies, respectively. The core challenge is that no embedding parameters exist for these unseen tokens, creating a fundamental representation gap. Common workarounds, such as mapping novel tokens to the nearest in-vocabulary value, introduce substantial approximation errors. Since financial time series are highly sensitive to exact values, these substitutions propagate inaccuracies through subsequent model layers, ultimately compromising prediction reliability. This representation failure severely hinders generalization, particularly during extreme market events (e.g., the negative oil price shock in 2020) when novel price regimes emerge.

### A.5 SMOTE and Sampling Strategy

SMOTE has been widely used in financial machine learning for handling imbalance (Ranjbaran et al., 2023; Li et al., 2023; Wei, 2025). In Section 4.4 we omitted the detailed discussion of SMOTE for brevity. Here, we provide a complete formulation of our oversampling strategy, clarify its role in our pipeline.

**Data Representation.** Let the training dataset consist of cluster-index token sequences,

$$\mathcal{D} = \{(x_i, y_i)\}_{i=1}^N, \qquad x_i \in \{1, 2, \dots, K\}^L, \ y_i \in \{1, 2, \dots, K\}.\tag{19}$$

where each $x_i$ is a symbolic sequence of cluster indices of length $L$, obtained from the price-pattern encoder. **SMOTE is applied exclusively to these discrete token vectors, not to raw or continuous price series.** This design completely avoids the risk of synthesizing economically implausible price trajectories. For each label $k$, let

$$n_k = \big|\{i : y_i = k\}\big| \quad \text{and} \quad n^* = \text{median}\{n_k\}_{k=1}^K.\tag{20}$$

The final number of samples per class is defined as

$$n_k^{\text{final}} = \begin{cases} n^*, & n_k = 1 \quad \text{(extremely rare class, duplicated)}, \\ n^*, & 2 \leq n_k < n^* \quad \text{(minority class, oversampled via SMOTE)}, \\ n^*, & n_k > n^* \quad \text{(majority class, downsampled)}. \end{cases} \quad (21)$$

**SMOTE in the Token Space.** For minority classes with $2 \leq n_k < n^*$, synthetic samples are generated using SMOTE applied to the *token vectors* $x_i$. Given a sample $x_i$ and one of its $K$ nearest neighbors $x_{nn}$ in the token space (we use $K = 2$ throughout),

$$\tilde{x} = x_i + \lambda(x_{nn} - x_i), \qquad \lambda \sim U(0, 1). \quad (22)$$

Since the space is discrete, $(x_{nn} - x_i)$ is computed elementwise and the result is rounded back to valid cluster indices. Critically, this interpolation occurs *only between existing cluster indices*, never between price values. Thus, SMOTE redistributes density in the symbolic state space without creating artificial price paths.

**Even if SMOTE were applied to continuous features,** its linear interpolation cannot produce outliers far from the data manifold. We verified this behavior visually: as shown in Fig. 8, when applied to the bullish cluster, the synthetic samples (cyan) remain tightly concentrated around the original cluster (green), indicating SMOTE's inherently local and non-distortive nature.

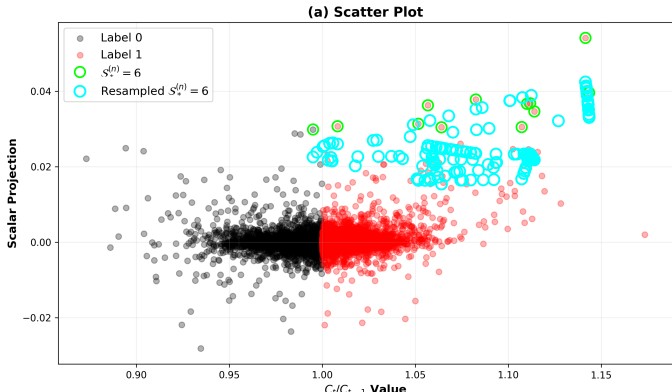

Figure 8: SMOTE applied to the bullish cluster: synthesized samples (cyan) tightly adhere to the original cluster boundary (green).

Table 12: Precision (%) and Counts Changes (%) without SMOTE

|  | S&P500 | | | | CSI300 | | | |
|---|---|---|---|---|---|---|---|---|
|  | Ours5 | | Ours10 | | Ours5 | | Ours10 | |
|  | Prec | Count | Prec | Count | Prec | Count | Prec | Count |
| 2011–2020 | +0.12 | -9% | +6.45 | -58% | +0.04 | -26% | -0.28 | -31% |
| 2015–2024 | -1.19 | -99% | — | -100% | -0.38 | -37% | +1.32 | -95% |

**Practical Role of SMOTE.** The SMOTE module substantially stabilizes our classifier. Without SMOTE, prediction counts decrease sharply (Table 12), leading to reduced robustness in detecting bullish-cluster events. The oversampling procedure restores class balance and yields more reliable bullish predictions across both markets.

Overall, SMOTE in our framework operates purely in the symbolic cluster-index space, enhances prediction robustness, and does not introduce any risk of generating unrealistic price dynamics.

## A.6 SELECTION OF OPTIMAL $K$ FOR BULLISH CLUSTERS

The weights in our scoring function (Eq. 10) were designed to balance three competing objectives: maximizing bullish probability, maintaining adequate cluster size, and controlling model complexity. The weight settings reflect our preferences derived from this foundational dataset. To illustrate this principle, we present below the example used in the manuscript for the S&P 500 (2004–2013), where for each candidate $K$ we evaluate three quantities for the most bullish cluster: (1) its sample size, (2) the bullish ratio (percentage of samples whose next-day closing price increases), and (3) the resulting score computed using Eq. (10).

Table 13: Cluster statistics and calculated scores for S&P500 (2004–2013)

| K | Size | Ratio | Score | K | Size | Ratio | Score |
|---|---|---|---|---|---|---|---|
| 5 | 65 | 0.69 | 15.64 | 13 | 12 | 0.92 | 17.07 |
| 6 | 32 | 0.72 | 15.55 | 14 | 12 | 0.92 | 16.87 |
| 7 | 15 | 0.80 | 15.93 | 15 | 12 | 0.92 | 16.67 |
| 8 | 15 | 0.80 | 15.73 | 16 | 12 | 0.92 | 16.47 |
| 9 | 15 | 0.80 | 15.53 | 17 | 12 | 0.92 | 16.27 |
| 10 | 14 | 0.79 | 14.89 | 18 | 6 | 1.00 | 14.73 |
| 11 | 12 | 0.92 | 17.47 | 19 | 6 | 1.00 | 14.53 |
| 12 | 12 | 0.92 | 17.27 | 20 | 1 | 1.00 | -19.00 |

Among these, **the bullish ratio is the most important criterion**. A ratio of $100\%$ is theoretically ideal, but the values $K = 18$–$20$—which achieve $100\%$—have extremely small cluster sizes (all below 10, and only one sample when $K = 20$), indicating that the clustering has become overly fragmented and unreliable. Hence these $K$ values are discarded immediately. The next-best group consists of $K = 11$–$17$, all with ratio $0.92$ and identical sizes. Within this group, our design principle favors the *smallest* $K$, because a smaller $K$ leads to a more compact token vocabulary and avoids excessive token proliferation. Therefore, $K = 11$ is selected as the optimal choice.

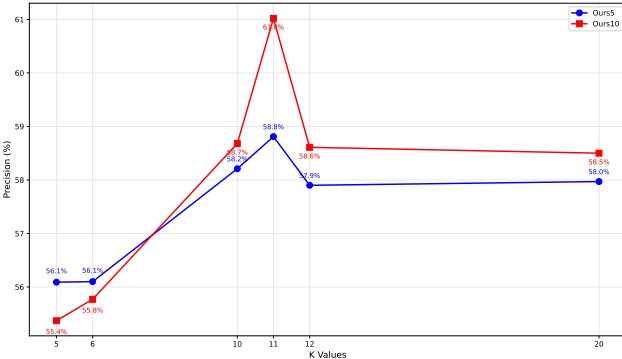

Figure 9: Precision comparison across different $K$ values (S&P500 2015–2024)

**Interpretation of the Weighting Scheme.** The weighting scheme in Eq. (10) is *explicitly constructed* to encode these principles:

- **Bullish ratio** receives the dominant positive weight (**+25.0**), because it is the primary determinant of cluster quality.
- **Cluster size** receives a negative weight (**-0.8**), penalizing clusters that become too small (e.g., $K = 18$–$20$).
- **The number of clusters** $K$ receives a mild negative weight (**-0.2**), reflecting our preference for a more compact token vocabulary.

Thus, the weights in Eq. (10) are not arbitrary—they directly implement the decision rule: *select the smallest $K$ that yields a stable, sufficiently large, and strongly bullish cluster.* As illustrated in Fig. 9, both extremely small ($K = 5, 6$) and excessively large ($K = 20$) values exhibit suboptimal

performance. When $K$ is too small, clusters show weak bullish predictive power. When $K$ is too large, clusters become unstable and statistically unreliable. Among the evaluated values ($K = 5, 6, 10, 11, 12, 20$), $K = 11$ achieves the highest precision, providing an optimal balance between stability and predictive accuracy.

**Additional Example: S&P500 (2001–2009).** The same logic applies when selecting the optimal $K$ for the S&P500 during 2001–2009. Even without explicitly computing the scores using Eq. (10), the choice is visually evident from Table 14: the extreme small clusters corresponding to $K = 17$–$20$ can be ignored, as they contain only 1–2 samples. Among the remaining clusters, $K = 6$ and $K = 7$ exhibit the highest ratio (0.824). Since we seek the **smallest** $K$ that satisfies stability and size criteria, $K = 6$ is naturally selected.

Table 14: Cluster statistics and calculated scores for S&P500 (2001–2009)

| K | Size | Ratio | Score | K | Size | Ratio | Score |
|---|------|-------|-------|---|------|-------|-------|
| 5 | 37 | 0.811 | 18.19 | 13 | 22 | 0.773 | 14.90 |
| 6 | 34 | 0.824 | 18.21 | 14 | 22 | 0.773 | 14.70 |
| 7 | 34 | 0.824 | 18.01 | 15 | 22 | 0.773 | 14.50 |
| 8 | 27 | 0.815 | 17.29 | 16 | 22 | 0.773 | 14.30 |
| 9 | 23 | 0.783 | 16.03 | 17 | 2 | 1.00 | 1.60 |
| 10 | 23 | 0.783 | 15.83 | 18 | 2 | 1.00 | 1.40 |
| 11 | 23 | 0.783 | 15.63 | 19 | 2 | 1.00 | 1.20 |
| 12 | 23 | 0.783 | 15.43 | 20 | 2 | 1.00 | 1.00 |

The weights in our scoring function (Eq. 10) should therefore be understood not as externally tuned hyperparameters, but as a concise summary of the decision principles described above. That is, the weighting scheme was distilled from the observed behaviors of different $K$ values across datasets: clusters with extremely small sizes must be penalized, overly large $K$ values must be discouraged, and the bullish ratio must dominate the evaluation. The final coefficients encode these empirically derived preferences in a compact mathematical form, ensuring consistent $K$-selection across all datasets.

A.7 PERFORMANCE ACROSS BASELINE MODELS

Table 15: Comparison of Our Tokenization Strategy and Baseline Models

| Method | Input | Output | Configuration |
|--------|-------|--------|---------------|
| Ours | 9 tokens | $\mathbb{I}(\hat{z}_i^{(10)} \in \mathcal{S}_*^{(10)})$ | Architecture: Refer to Section 5.2. |
| RF | 36 features | $\mathbb{I}(\hat{C}_t/C_{t-1} > \tau)$ | 150 estimators; Unlimited depth; Min split 2; Min leaf 2; Max features 6. |
| XGB | 36 features | $\mathbb{I}(\hat{C}_t/C_{t-1} > \tau)$ | 100 estimators; Max depth 5; $\eta = 0.1$; Subsample 1; Column sample by tree 0.8. |
| LSTM | 36 sequences | $\mathbb{I}(\hat{C}_t/C_{t-1} > \tau)$ | Two layers, 64 units ($\eta = 0.001$); First returns sequences; Second to output; Dropout 0.3. |
| CNN | 6x6 matrix | $\mathbb{I}(\hat{C}_t/C_{t-1} > \tau)$ | Convolutional layers(64,128)+pooling; Flatten; Dense-256; Dropout 0.1. |
| SVM | 36 features | $\mathbb{I}(\hat{C}_t/C_{t-1} > \tau)$ | Linear kernel; The regularization parameter $C$ = 1 with automatic gamma scaling. |
| Lasso | 36 features | $\mathbb{I}(\hat{C}_t/C_{t-1} > \tau)$ | $\ell_1$ regularization with strength $\alpha = 1.0$; 5000 iterations |

In Table 15 we also evaluate our tokenization method alongside six established machine learning models—Random Forest (RF), Extreme Gradient Boosting (XGBoost), Long Short-Term Memory (LSTM), Convolutional Neural Network (CNN), Support Vector Machine (SVM), and Lasso Regression (Dubey et al., 2024)—within a unified experimental framework. All baseline methods also use the same normalization procedure (division by $C_{t-1}$) as global data statistics are unavailable in

practice. Each model is trained to predict $C_t/C_{t-1}$, and predictions are thresholded at four levels ($\tau \in [1.00, 1.03]$) to assess sensitivity, with the prediction defined as $\hat{y} = \mathbb{I}(C_t/C_{t-1} > \tau)$.

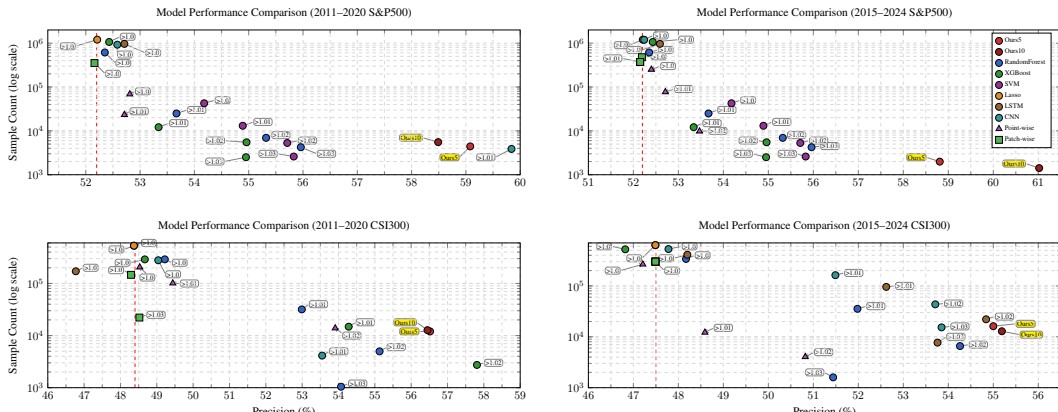

Figure 10: Model Performance Comparison on constituent stocks of S&P500 and CSI300 indices. The graphs show the precision of various models for the periods 2011–2020 and 2015–2024.

Unlike traditional threshold-dependent models that require careful selection of optimal $\tau$ values, our method requires no predefined $\tau$ and attains high precision directly from clustered tokens. The Transformer architecture equipped with our multi-scale feature extraction and eigencluster-based tokenization method demonstrates competitive performance that matches or surpasses conventional benchmarks across most test conditions. Our best model achieves 61% precision on 2015–2024 S&P500, which translates to successfully predicting price increases in 61 out of every 100 forecasts. While some traditional models (e.g., CNN at $\tau > 1.01$ on S&P 500 and XGBoost at $\tau > 1.02$ on CSI 300) show narrow-range competitiveness, such advantages are practically negligible since optimal $\tau$ values are unknown in real trading scenarios. This underscores the robustness and practical superiority of our $\tau$-free tokenization strategy.

---

**Algorithm 1** Cluster-Enhanced Trading Strategy (Adjusted Timing)

---

**Setting:** Initial capital \$50,000, transaction fee 1%
1: Allocate 85% (\$42,500) to S&P 500 at $t_0$ (close price $C_{t_0}^{\text{S\&P}}$)
2: Reserve 15% (\$7,500) as active capital; initialize Profit = 0
3: **for** each trading day $t_i$, $i \geq 1$ **do**
4:      Previous day $t_{i-1}$:
5:      **if** model predicts $N$ stocks to buy **then**
6:          Allocation$_j$ = \$7,500/N$
7:          **for** each stock $j$ **do**
8:              $S_j = \lfloor \text{Allocation}_j/C_{t_{i-1},j} \rfloor$; Buy $S_j$ shares at $C_{t_{i-1},j}$
9:          **end for**
10:     **end if**
11:     Today $t_i$:
12:     **if** held stocks exist **then**
13:         **for** each stock $j$ **do**
14:             Sell $S_j$ shares at $C_{t_i,j}$
15:             Profit += $S_j \times (C_{t_i,j} - C_{t_{i-1},j}) \times 0.99$
16:         **end for**
17:     **end if**
18:     Update index value: IndexValue = (\$42,500/C_{t_0}^{\text{S\&P}}) \times C_{t_i}^{\text{S\&P}}$
19: **end for**
**Output:** Final portfolio value = IndexValue + Profit

---

## A.8   TRADING BACKTEST

To evaluate the practical utility of our cluster-enriched transformer predictions, we conducted a trading backtest on the US stock market from 2015 to 2024 using the `Ours5` model. The strategy, summarized in Algorithm 1, maintains a core allocation to the S&P 500 index (85% of total capital) while deploying the remaining 15% for active daily trading based on the model's predictions. All selected stocks are equally weighted, positions are liquidated the following trading day to capture short-term cluster effects, and a transaction cost of 1% per trade is assumed to reflect realistic execution frictions. The performance is quantified using the *Total Return (TotR)* metric:

$$\text{TotR} = \frac{P_T - P_0}{P_0}, \tag{23}$$

where $P_0 = \$50{,}000$ is the initial capital and $P_T$ is the terminal portfolio value.

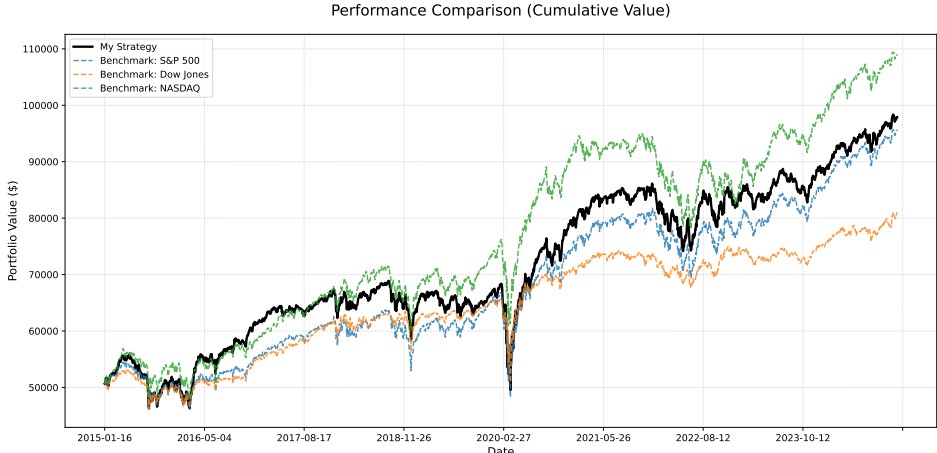

Figure 11: Backtest performance comparison of `Ours5` model trading strategy (2015–2024) against benchmark indices.

Our `Ours5`-based trading strategy achieved a total return of 95.8% from 2015 to 2024, outperforming the S&P 500 (91.4%) and the Dow Jones Industrial Average (62.1%), while the Nasdaq Composite led the benchmarks with 118.1%. As shown in Figure 11, our strategy consistently outperforms the buy-and-hold approach on the S&P 500 and Dow Jones indices, while the Nasdaq Composite slightly surpasses our returns, reflecting the strong performance of US technology stocks. These results demonstrate that the higher precision of our model in predicting bullish signals can be effectively translated into tangible trading gains beyond passive index investing.

## A.9   TABULATED RESULTS CORRESPONDING TO FIG. 6

Table 16: Precision Comparison (%) of Tokenization Methods

| Index | Period | Point-wise (3 decimal) | | | | Patch-wise | | | | Ours | |
|-------|--------|-------|--------|--------|--------|-------|--------|--------|--------|------|------|
| | | $> 1.0$ | $> 1.01$ | $> 1.02$ | $> 1.03$ | $> 1.0$ | $> 1.01$ | $> 1.02$ | $> 1.03$ | 5 | 10 |
| S&P500 | 11-20 | 52.81 | 52.71 | 50.29 | 53.70 | 52.16 | 52.16 | 52.16 | 52.16 | 59.08 | 58.49 |
| | 15-24 | 52.40 | 52.71 | 53.47 | 53.91 | 52.19 | 52.15 | 52.15 | 52.15 | 58.81 | 61.02 |
| CSI300 | 11-20 | 48.53 | 49.44 | 53.91 | 52.58 | 48.29 | 48.29 | 48.29 | 48.52 | 56.52 | 56.45 |
| | 15-24 | 47.21 | 48.59 | 50.82 | 49.85 | 47.49 | 47.49 | 47.49 | 47.49 | 55.00 | 55.19 |

## A.10   LLM USAGE

The authors primarily used Grok4, ChatGPT, and Deepseek for linguistic polishing and paragraph refinement only. All aspects of coding, research, analysis, figure preparation, and manuscript composition were performed solely by the authors without AI assistance.

