# OpenReview forum: "Learn Bullish Moves via EigenCluster Tokens"
_ICLR.cc/2026/Conference — Submitted to ICLR 2026_

### Official Review · Reviewer_a6aw · 2025-10-24

**Soundness:** 3
**Presentation:** 3
**Contribution:** 2
**Rating:** 4
**Confidence:** 4

**Summary:**

This paper proposes EigenCluster Tokens, a clustering-based tokenization method for financial time series that uses eigendecomposition and multi-scale analysis to generate compact, meaningful tokens. It enables Transformers to predict bullish movements with 6–9% higher precision than market baselines, reduces token sequence length by 75%, and achieves superior efficiency and real-world trading performance.

**Strengths:**

+ Originality: Proposes a novel, domain-specific tokenization paradigm—EigenCluster Tokens—by combining spectral decomposition and clustering for financial time series, reframing prediction as bullish cluster identification.
+ Quality: Methodologically rigorous with sound mathematical formulation and thorough experiments across multiple datasets, baselines, and ablation settings.
+ Clarity: Well-structured and clearly written, with intuitive visualizations (e.g., multi-scale workflow, cluster separation) and logical progression from problem to solution.
+ Significance: Highlights tokenization—not architecture—as the key bottleneck in financial forecasting with Transformers. Offers practical benefits: compact representations, faster inference, and actionable trading signals.

**Weaknesses:**

+ Lack of Justification for Key Methodological Choices:
  + Arbitrary Scalar Projection Function: The function to project high-dimensional features into a scalar (Eq. 6), using a specific combination of $sin()$ and $L2$-norm, lacks theoretical or intuitive explanation. It is unclear why this complex form is superior to simpler, more interpretable alternatives (e.g., using the principal components directly).
  + Counter-intuitive "Bullish Cluster" Criterion: The rule for selecting the "bullish cluster" by favoring the "smallest" size (Eq. 9) is counter-intuitive. This could lead the model to overfit to a few unrepresentative outliers. A more robust criterion would balance the cluster's size with its predictive precision.
+ Insufficient and Potentially Misleading Comparisons:
  + Weak Baseline Models: The comparison is primarily against many weak baseline (e.g. XGBoost, LSTM, Lasso), which are not state-of-the-art baselines. A more convincing evaluation would involve comparing against established time series models like PatchTST or Autoformer.
  + Overly Simplistic Financial Backtest: The trading backtest is insufficient as it only compares against market indices. It should include stronger quantitative baselines (e.g., Time-Series Momentum) and report standard risk-adjusted performance metrics like the Sharpe Ratio and Maximum Drawdown.
+ Serious Concerns about Generalizability:
  + Hand-Tuned Hyperparameters: The formula for selecting the optimal number of clusters $K$ (Eq. 10) and its weights were empirically hand-tuned on a single dataset. This is a classic case of "hyperparameter fitting" that severely undermines the method's credibility and claims of generalization, making it feel more like a custom solution than a general framework.
  + Lack of Online Applicability: The entire tokenization process is performed offline. The paper fails to discuss how the method would adapt to continuously arriving new data in a real-world, online trading scenario, which is a critical practical omission.

**Questions:**

1. My primary concern is the justification for using a deep 8-layer Transformer with an extremely short input sequence of 9 tokens, trained on a limited set of index data. This setup raises significant questions about overfitting and the actual utility of a complex self-attention architecture. To address this, please **provide plots of the training and validation loss curves** from your experiments to demonstrate that the model is learning generalizable patterns rather than memorizing the training data. Additionally, please provide a clear justification—ideally supported by a comparative experiment with a simpler model like an LSTM—for why a deep Transformer is necessary and effective in this short-sequence, low-data regime.
2. The question to addresses the computational scalability of the multi-scale tokenization. The proposed method requires $n$ separate eigendecompositions on increasingly large matrices, which seems computationally prohibitive for the longer input sequences (e.g., n > 96) common in time series forecasting. To assess the practical viability of your approach, could you please provide both a formal complexity analysis and an empirical study showing **how the tokenization time, training time, and model precision scale as the sequence length $n$ increases**? This is crucial to understand whether the performance benefits justify the potentially exponential growth in computational cost.
3. The concern about the validity of the baseline comparisons. The paper compares its method against non-standard, discretized versions of *point-wise* and *patch-wise* tokenization. The standard approach in modern time series Transformers is to directly project continuous patches into the embedding space via a linear layer, which is a stronger baseline. To properly validate the contribution, the authors must add an experiment comparing their method against this standard **linear projection baseline** under the same model architecture. Without this, the paper's claims of superiority are not sufficiently substantiated, as it has not competed against the most common and robust alternative.

---

> ### Author Response · Authors · 2025-11-18
> **Linear Projection Baseline**
>
> We appreciate the reviewer’s suggestion to compare with a linear projection baseline. We understand that this concern pertains to **the design of the embedding layer.** Currently, there are [two major paradigms](https://github.com/MasterBeard/EigenCluster-Tokenization-for-Financial-Transformers/blob/main/motivation-2_01.jpg "Two paradigms: token embedding vs. linear projection") (click to view) for input embedding (as summarized in recent research by **Z. Liang et al., 2025**: (1) the traditional NLP-style token embedding (discrete token embedding with a large lookup table), and (2) the linear or convolutional embedding approaches adopted by models such as PatchTST and Autoformer (with shared weights that remove token-wise independence).
>
> **Our work specifically focuses on exploring how to design effective financial tokens under the NLP-style embedding paradigm for Transformer architectures**. This is because only within the NLP embedding setting does the concept of a token naturally arise—once linear or convolutional embeddings are used, the notion of tokenization essentially disappears, and the input representation becomes continuous rather than discrete tokens.
>
> ---
>
> **Related Work**
> - Liang, Z., Zhu, J., Sun, W. (2025). *Why attention fails: The degeneration of transformers into MLPs in time series forecasting*. arXiv:2509.20942.
> - Wu, H., Xu, J., Wang, J., Long, M. (2021). *Autoformer: Decomposition transformers with auto-correlation for long-term series forecasting*. NeurIPS.
> - Nie, Y., Nguyen, N. H., Sinthong, P., Kalagnanam, J. (2023). *A time series is worth 64 words: Long-term forecasting with transformers*. ICLR.
>
> ---
>
> While this concern is somewhat outside the main scope of our study—which focuses on token design rather than embedding projection mechanisms—we appreciate the reviewer’s suggestion and acknowledge that such a comparison may provide broader insight. Therefore, to further strengthen our claims, we conducted additional comparative experiments using different embedding methods while keeping the overall architecture unchanged. The only modification was replacing our NLP-style embedding layer with linear and convolutional mappings, implemented as follows:
>
> - **Linear embedding (PatchTST-style):**
> `nn.Linear(36, d_model)`
>
> - **Convolutional embedding (Autoformer-style):**
> `nn.Conv1d(in_channels=36, out_channels=d_model, kernel_size=3, padding=1, padding_mode='circular', bias=False)`
>
> We evaluated these variants on the S\&P 500 dataset (2015--2024 test set). As shown below, when the threshold exceeded \(1.02\), both the linear and convolutional embedding models failed to produce valid predictions. Even under thresholds \(>1.0\) and \(>1.01\), their performance remained consistently lower than our token-based models (Ours-5 and Ours-10).
>
> | Threshold | Linear Embedding | Convolutional Embedding |
> |-----------|------------------|--------------------------|
> | \(>1.0\)  | 52.54\%          | 52.66\%                 |
> | \(>1.01\) | 51.09\%          | 54.20\%                 |
> | \(>1.02\) | /                | /                       |
> | \(>1.03\) | /                | /                       |
> | **Ours-5**  | **58.81\%**     | **58.81\%**             |
> | **Ours-10** | **61.02\%**     | **61.02\%**             |

---

> ### Author Response · Authors · 2025-11-18
> **Longer Sequences (Model Precision Scale As The Sequence Length n Increases)**
>
> We appreciate the reviewer’s suggestion. For our task of *daily financial price forecasting*, we have carefully considered the issue of input length. **In the existing literature, models typically use the past one or two weeks of daily prices (around 5–10 tokens) as input (e.g., C. Wang et al., 2022; Q. Zhang et al., 2022).** Using a much longer context (e.g., 96 tokens, corresponding to about three months of history in our study) is uncommon in financial forecasting tasks, since market dynamics and volatility change rapidly over time. Empirically, distant historical information tends to have limited predictive value for short-term price movements, and including it may even introduce noise rather than improving forecasting accuracy.
>
> To further address the reviewers’ concerns, we conducted additional experiments comparing input lengths of 5, 10, 15, and 30 tokens. The experiments were conducted on the S&P 500 dataset covering the period from 2015 to 2024. The results are shown below:
>
> | Total Tokens | Input Tokens | Predicted Tokens | Precision (%) |
> |--------------|-------------|----------------|---------------|
> | 5            | 4           | 1              | **61.99**     |
> | 10           | 9           | 1              | 61.02         |
> | 15           | 14          | 1              | 54.94         |
> | 30           | 29          | 1              | 52.92         |
>
> The results indicate that increasing the input length does not necessarily improve performance; in fact, longer input sequences tend to harm the model’s ability to identify next-day upward trends. Conversely, shorter input lengths (e.g., 4 historical tokens predicting the next one) are more effective for short-term trend forecasting.
>
> In terms of computation time, the difference is minimal. On the same Colab T4 GPU, using a batch size of 512, the test set takes about 45 seconds for 5 input tokens and 49 seconds for 30 input tokens. The tokenization and training times are almost identical, since the training dataset is much smaller than the test set, and these steps are not significantly affected by input length.
>
> **References**
>
> - Wang, Chaojie, Chen, Yuanyuan, Zhang, Shuqi, & Zhang, Qiuhui (2022). *Stock market index prediction using deep transformer model*. Expert Systems with Applications, 208, 118128.
>
> - Zhang, Qiuyue, Qin, Chao, Zhang, Yunfeng, Bao, Fangxun, Zhang, Caiming, & Liu, Peide (2022). *Transformer-based attention network for stock movement prediction*. Expert Systems with Applications, 202, 117239.

---

> ### Author Response · Authors · 2025-11-18
> **Loss Curves and Depth Effects**
>
> Thank you for raising this important point. Regarding the concern that an 8-layer Transformer might overfit when applied to an input sequence of length 9 with limited training data, we provide the following clarifications and additional evidence:
>
> 1. **Meaning of the 9 input tokens**
>    The “9 tokens” mentioned in the manuscript are not 9 raw observations. Instead, each token represents a multi-scale aggregate derived from the original 180 OHLC values, corresponding to window sizes `[36, 32, 28, 24, 20, 16, 12, 8, 4]`. These aggregated features encode multi-resolution statistical structure and cross-scale dependencies. Thus, although the sequence length is short, each token contains rich high-dimensional information.
>
> 2. **Training and validation learning dynamics**
>    We have included the training and validation [loss curves](https://github.com/MasterBeard/EigenCluster-Tokenization-for-Financial-Transformers/blob/main/loss_curve_01.jpg "Loss Curves") (click to view). Both training and validation losses consistently decrease throughout the entire training process, with no late-stage increase in validation loss. This indicates that the model is learning generalizable patterns rather than memorizing the training data.
>
> 3. **Regarding the choice of an 8-layer Transformer**
>    This choice was made for two practical reasons:
>    - The point-wise tokenization produces the longest sequence (36 tokens). With a fixed hidden size (`d=64`), a very shallow Transformer would underfit longer sequences, leading to unfair comparison across tokenization schemes.
>    - The depth aligns with common practice in prior work, where architectures such as the original Transformer and Informer typically apply 6–10 layers (encoder + decoder). With `d=64`, eight layers remains a moderate configuration.
>
> **Again we clarify that the goal of this work is not to argue that a deep Transformer is essential for short-sequence, low-data financial prediction, nor that it outperforms other model families such as LSTMs.** Our contribution focuses on a narrower and controlled research question:
>
> > *Given the same vanilla Transformer backbone and NLP-style embedding setup, which tokenization scheme is most effective for short-term upward movement prediction?*
>
> To isolate the effect of tokenization, all architectural and hyperparameter components are intentionally kept fixed. Therefore, the depth serves as a consistent experimental backbone rather than being optimized.
>
> Although architectural comparison is beyond the main scope, we conducted an additional test to address the reviewer’s concern. We compared the original 8-layer Transformer with a shallower 2-layer variant and a 1-layer LSTM baseline on the S&P 500 (2015–2024) test set. Results (precision) are shown below:
>
> | Model                     | Precision (%) |
> |----------------------------|---------------|
> | 8-layer Transformer (Ours-10) | **61.02**     |
> | 2-layer Transformer (Ours-10) | 59.63         |
> | 1-layer LSTM               | 54.86         |
>
> These results suggest that, at least on this dataset, a deeper model offers moderate performance gains over shallower variants and clearly outperforms the LSTM baseline. Further architectural comparisons remain an interesting direction for future work.

---

> ### Author Response · Authors · 2025-11-18
> **“Bullish Cluster” Criterion and Weak Baseline Models**
>
> **A.“Bullish Cluster” Criterion (@Reviewer a6aw)**
>
> We respectfully note that the so-called “counter-intuitive” nature of our Bullish Cluster criterion arises from a misunderstanding of the underlying dynamics. In fact, the most bullish cluster is naturally expected to have a small size.
>
> - Clusters closer to the overall distribution center are larger and exhibit roughly balanced proportions of upward and downward movements, reflecting the majority of neutral or typical behavior.
> - Clusters farther from the center are smaller but have higher upward-movement ratios, which aligns with our criterion for identifying bullish clusters.
>
> This is illustrated clearly in Figure 4 of the main text. Therefore, it is inherently unlikely to observe a cluster that simultaneously has a large size, a high upward probability, and remains near the distribution center. The Bullish Cluster criterion is consistent with the intuitive behavior of extreme upward trajectories in the data.
>
> **B.Weak Baseline Models ($@$Reviewer a6aw)**
>
> The reviewer also raised a concern that our selected baselines (XGBoost, LSTM, and Lasso) are “weak” and would like to see the comparison with PatchTST and Autoformer.  We would like to clarify that these models are not weak; rather, they are well-established and widely used in financial forecasting (e.g., Dubey et al., 2024; Thakkar & Chaudhari, 2021). Their “age” does not imply inferior performance. We would like to provide the following clarifications:
>
> 1. **Focus on NLP-style embeddings and tokenization**
>    PatchTST and Autoformer are designed for time series forecasting, but their embedding mechanisms differ from traditional NLP embeddings. In other words, these models do not operate on “tokens” in the NLP sense. Our study specifically investigates which tokenization schemes work best under a standard NLP-style embedding. Directly comparing against PatchTST or Autoformer would conflate the effects of token design with the effects of embedding architecture, making it impossible to isolate the benefit of tokenization.
>
> 2. **Practical limitations of reproducing PatchTST**
>    We attempted to reproduce PatchTST using the [NeuralForecast](https://nixtlaverse.nixtla.io/neuralforecast/docs/getting-started/introduction.html) package. However, a direct comparison is infeasible due to the training paradigm: PatchTST trains on a long sequence and predicts the next sequence, minimizing MSE over the predicted segment. **To predict subsequent segments, it requires retraining on each new input sequence.** Our test set includes nearly 500 stocks with approximately 2,500 trading days each (~2,500 sequences per stock), resulting in an estimated 1.25 million separate training runs just for one of test sets—well beyond current computational feasibility.
>
> 3. **Empirical observations from prior work**
>    In particular, recent work by Zeng et al. (2023) shows that these Transformer-based models for time series forecasting, such as Informer or Autoformer, can even underperform simple linear or classical baselines. This indicates that these TS-Transformer models do not automatically guarantee superior performance in practice.
>
> **References**
>
> - Dubey, A., Singh, S., Mishra, A. K., et al. (2024). *A survey on machine learning techniques for stock market price prediction*. In Proc. 2024 ASU Int. Conf. Emerging Technologies for Sustainability and Intelligent Systems (ICETSIS), pp. 682–691.
> - Thakkar, A., & Chaudhari, K. (2021). *A comprehensive survey on deep neural networks for stock market: The need, challenges, and future directions*. Expert Systems with Applications, 177, 114800.
> - Zeng, A., Chen, M., Zhang, L., & Xu, Q. (2023). *Are transformers effective for time series forecasting?* In *Proceedings of the AAAI Conference on Artificial Intelligence (AAAI'23)*.
>
> **C.Lack of Online Applicability (@Reviewer a6aw)**
>
> We respectfully note that while our methodology and experiments are fully open-sourced and can be reproduced using the provided historical data and code, we cannot disclose details of our live trading operations. These operations constitute proprietary trading strategies, and revealing them would compromise commercial confidentiality. Nonetheless, the results on historical data are fully reproducible, ensuring that the scientific contributions of this work remain verifiable.

---

> ### Author Response · Authors · 2025-11-18
> **Scalar Projection and The Formula for Selecting K**
>
> Regarding the Scalar Projection function, please refer to our detailed response to @Reviewer U9Fv. Briefly, the Scalar Projection is primarily intended to facilitate visual explanation. Our audience also includes traders and investors, for whom interpreting multi-dimensional clustering directly is impractical. By projecting clusters onto a single dimension, the results become immediately understandable without altering the underlying clustering structure.
>
> As for the formula used to select the optimal number of clusters, we refer the reviewer to our response to @Reviewer CR4w, where we explain the rationale and criteria for determining the optimal \(K\).

---

### Official Review · Reviewer_U9Fv · 2025-10-29

**Soundness:** 2
**Presentation:** 2
**Contribution:** 2
**Rating:** 4
**Confidence:** 5

**Summary:**

The paper proposes eigencluster tokenization, a multi-scale spectral method that discretizes OHLC time series into semantic tokens for Transformer-based next-day up-move prediction. It achieves higher precision and faster inference than point- or patch-wise baselines, with complementary gains from multi-scale design, eigendecomposition, and clustering.

**Strengths:**

1. The paper identifies a clear and significant research problem, focusing on tokenization and out-of-vocabulary (OOV) challenges in financial time-series modeling, which are crucial for improving model generalization and interpretability.

2. The proposed approach shows good engineering feasibility, as the multi-scale prefix, eigendecomposition, and clustering pipeline forms a concise and implementable framework that reduces token and sequence length while preserving key temporal information.

3. The authors exhibit commendable efforts toward reproducibility by providing code access and detailed implementation descriptions in the appendix, which facilitates transparent verification and future research replication.

**Weaknesses:**

1. The current scalar projection design lacks theoretical grounding and comparative evaluation. It is recommended to test alternative mappings (e.g., first principal component, eigenvector projection, multi-component clustering, or nonlinear functions like tanh) and report robustness across settings.

2. In terms of clustering methodology, automatic selection approaches such as the Bayesian Information Criterion, Akaike Information Criterion, Bayesian Gaussian Mixture Models, or time-series cross-validation should be compared. The paper should also justify the rationale and advantages of the specific clustering method it adopts.

3. There already exist other clustering-based tokenization methods, but the paper does not cite or compare them to demonstrate the advantages of its proposed tokenization approach.

4. The paper does not provide sufficient justification for the effectiveness of the chosen prefix windowing approach, and the selection of the prefix window size lacks rationale and comparative experiments.

5. The baselines include only point- and patch-wise discretization, but not learned tokenizers such as VQ-VAE, discrete autoencoders, or transformer-based quantizers, which would provide a stronger benchmark for the proposed approach.

6. The paper lacks a principled discussion of why eigen-decomposition and 1D clustering should preserve predictive structure or how they relate to temporal-spectral representations.

**Questions:**

1) What is the theoretical motivation for the specific scalar projection design? Have alternative mappings or nonlinear transformations been tested to confirm robustness?

2) What is the principled link between eigendecomposition + 1D clustering and the preservation of predictive temporal–spectral structure in price series?

3) Why was 1D K-means chosen over automatic or probabilistic clustering methods (e.g., BIC/AIC-based selection or Bayesian GMM)? How sensitive are the results to this choice?

---

> ### Author Response · Authors · 2025-11-18
> **Scalar Projection Design and K-means Chosen and The Principled Link**
>
> We thank the reviewer for their careful review. In this response, we focus on addressing three main points raised:
>
> **A. Scalar Projection Design (@Reviewer U9Fv)**
>
> We clarify that the scalar projection step is applied **before clustering**, but it does **not alter the clustering results**. The transformation is constructed as a *monotonic mapping* of the projected feature representations, which preserves the relative ordering and geometry relevant to cluster assignment. Consequently, applying or removing this step yields identical clustering outcomes.
>
> The use of $\sin(\cdot)$ provides bounded signed directional information, while the Euclidean norm summarizes magnitude. The resulting scalar therefore reflects a direction–magnitude composite that is interpretable, yet structure-preserving. In contrast, alternatives such as using the first principal component or nonlinear transformations (e.g., kernel mappings, t-SNE, UMAP) would modify the underlying feature geometry and therefore *would change* clustering results — which is intentionally avoided.
>
> **The motivation for introducing this projection is interpretability rather than algorithmic necessity.** It is solely intended for visualization and to help interpret what the clusters look like. It enables a consistent one-dimensional ordering of clusters, providing a clear semantic axis for communication (e.g., bearish → neutral → bullish). Without such a representation, interpreting or explaining a cluster in 40+ dimensions is impractical, particularly for practitioner audiences such as investors and traders.
>
> We will update the manuscript to explicitly state this role and its invariance with respect to clustering outcomes.
>
> **B.1D K-means chosen (@Reviewer U9Fv)**
>
> We tested various automatic clustering approaches on the SP500 training set (2004-2013), including GMM with AIC/BIC selection and Bayesian GMM. The bullish cluster probabilities for these methods indicate the proportion of samples within the cluster whose next-day price increased, and these probabilities were only moderate (75\%, 68\%, and 61\%, respectively):
>
> | Clustering Method   | Selection | Bullish Cluster (%) |
> |--------------------|-----------|------------------|
> | GMM                | AIC       | 75%              |
> | GMM                | BIC       | 68%              |
> | Bayesian GMM       | Default   | 61%              |
> | K-means            | k=11      | **92%**          |
>
> Using 1D K-means with \(k=11\) produced a top cluster with a bullish probability of 92%, substantially stronger than the alternatives. Therefore, we chose 1D K-means as our main clustering method to obtain the most informative and actionable bullish cluster.
>
> **C.The principled link between eigendecomposition and 1D clustering (@Reviewer U9Fv)**
>
> In our study, the eigendecomposition step is effectively a PCA operation, but unlike conventional PCA, all eigenvectors are preserved instead of selecting only the leading components. PCA can enhance K-means clustering because it maximizes the variance of the data, making the cluster directions more distinct. For example, Ding and He (2004) demonstrated that principal components provide continuous solutions corresponding to discrete cluster membership indicators. Similarly, Zha et al. (2001) showed that a relaxed version of the K-means objective can be solved via partial eigendecomposition of the data Gram matrix, linking spectral methods with cluster assignment.
>
> **References**
>
> - Ding, C., & He, X. (2004). *K-means clustering via principal component analysis*. ICML.
> - Zha, H., He, X., Ding, C., Simon, H., & Gu, M. (2001). *Spectral relaxation for K-means clustering*. NIPS.

---

> ### Author Response · Authors · 2025-11-18
> **Limitations of Discrete Autoencoders and VQ-VAE  on Normalized Financial Data and Prefix Window Rationale**
>
> **Comparison with other methods (VQ-VAE, discrete autoencoders)**
>
> We thank the reviewer for the suggestion. We actually experimented with VQ-VAE and discrete autoencoders as alternative tokenization approaches; however, these methods collapse when applied to normalized financial price series, where values concentrate near the mean with very low variance. As a result, the quantization stage maps almost all sequences to one identical discrete token, producing no meaningful token diversity.
>
> Our training data consist of normalized financial features with extremely low variance:
> \begin{equation}
> x_{i,j} \in \mathbb{R}, \qquad
> \mu \approx 1.0005, \;\; \sigma \approx 0.0328, \qquad
> x_{i,j} = \mu + \varepsilon_{i,j}, \quad |\varepsilon_{i,j}| \ll 1,
> \end{equation}
> where $x_{i,j}$ denotes the $j$-th feature of the $i$-th sample, $\mu$ is the empirical mean, $\sigma$ is the empirical standard deviation, and $\varepsilon_{i,j}$ represents the small deviation from the mean.
>
> For any encoder $f_\theta$ (in VQ-VAE or a discrete autoencoder), such nearly constant inputs produce almost identical latent vectors:
> \begin{equation}
> z_i = f_\theta(x_i) \approx z_j, \quad \forall i,j,
> \end{equation}
> where $z_i \in \mathbb{R}^d$ is the latent representation of sample $i$.
>
> Consequently, the quantization layer (e.g., Gumbel-Softmax or VQ codebook) receives nearly identical logits for each sample:
> \begin{equation}
> \ell_i = W z_i + b \approx \ell_j, \quad
> \arg\max_k \ell_{i,k} = k_0, \quad \text{for all } i,
> \end{equation}
> where $W \in \mathbb{R}^{K \times d}$ and $b \in \mathbb{R}^K$ are learnable parameters projecting $z_i$ to $K$ logits, $\ell_i \in \mathbb{R}^K$ is the vector of logits for the codebook entries, $\ell_{i,k}$ denotes the $k$-th component of $\ell_i$, and $k_0$ is the single code index assigned to all samples.
>
> Thus, the discrete autoencoder collapses to a single token—not because of a model failure, but due to the low-variance, unimodal structure of normalized financial data, which lacks the diversity needed for meaningful discrete token learning.
>
> For transparency and reproducibility, we have uploaded a Colab notebook to an anonymous GitHub repository. The notebook contains the VQ-VAE/discrete autoencoder experiments and can be run directly on Colab to verify the behavior: [GitHub link](https://github.com/MasterBeard/EigenCluster-Tokenization-for-Financial-Transformers/blob/main/ICLR_review3_VQ_VAE.ipynb).
>
> **The selection of the prefix window size lacks rationale and comparative experiments (@Reviewer U9Fv)**
>
> We thank the reviewer for this comment. In our design, the prefix window grows proportionally with the sliding window length. Since each trading day contains OHLC features, longer prefixes naturally incorporate a wider span of historical volatility and market structure. Specifically, our chosen configuration:
>
> $$
> 40, 36, 32, 28, 24, 20, 16, 12, 8, 4
> $$
>
> is motivated by financial intuition: longer windows capture broader market regimes and smoother trends, whereas shorter windows emphasize localized price variations and short-term fluctuations. This approach is consistent with the design rationale in TimeMixer (Wang et al., ICLR 2024), where prefix lengths scale with the sliding window to align longer patterns with proportionally extended historical context.
>
> Alternative schedules, such as a coarser progression
>
> $$
> 40, 32, 24, 16, 8
> $$
>
> are less granular and may fail to capture intermediate temporal scales. Out of respect for the reviewer’s suggestion, we conducted an additional experiment comparing:
>
> - **Fine-grained schedule (used in the paper):**
>   $$40, 36, 32, 28, 24, 20, 16, 12, 8, 4$$
>
> - **Coarse schedule:**
>   $$40, 32, 24, 16, 8$$
>
> Evaluated on the **S&P500 (2011–2020)** dataset, the resulting precision scores are:
>
> | Schedule Type           | Precision |
> |-------------------------|-----------|
> | Fine-grained (10 steps) | **58.49%** |
> | Coarse (5 steps)        | 57.02% |
>
> The fine-grained schedule provides a modest improvement (+1.47%), supporting our intuition that more granular prefixes preserve information at intermediate temporal scales. Nonetheless, we acknowledge that a systematic evaluation of different prefix schedules could offer further insights into model sensitivity, and we plan to explore this in future work.
>
> **Reference**
>
> - Wang, S., Wu, H., Shi, X., Hu, T., Luo, H., Ma, L., Zhang, J. Y., & Zhou, J. (2024). *Timemixer: Decomposable multiscale mixing for time series forecasting*. International Conference on Learning Representations (ICLR).

---

### Official Review · Reviewer_18H9 · 2025-11-01

**Soundness:** 3
**Presentation:** 2
**Contribution:** 2
**Rating:** 6
**Confidence:** 4

**Summary:**

This paper proposes a new financial time series discretization method, Eigencluster Tokenization, which performs multi-scale eigendecomposition on OHLC matrices and clusters eigenvector projections to generate compact and semantically meaningful tokens. The approach aims to overcome three major issues in conventional point-wise and patch-wise tokenization for financial data: excessive token counts, sparse distributions, and out-of-vocabulary (OOV) risks. The authors conduct cross-market experiments on both S&P 500 and CSI 300 datasets.

**Strengths:**

1. The paper clearly discusses the challenges of financial tokenization that differ from language modeling (e.g., high variability, OOV risk), and proposes a multi-scale eigendecomposition and clustering approach on OHLC matrices. Made adaptations specific to financial data characteristics, such as using C_t−n+k−1 for normalization.


2. The definition of the most bullish cluster enhances model interpretability.

3. The evaluation on both S&P 500 and CSI 300 datasets demonstrates strong performance, along with significant reductions in token count and computational cost.

**Weaknesses:**

1. The explanation of Table 1 is unclear.

2. The clustering method is limited to K-means only; the impact of alternative clustering approaches (e.g., GMM, Spectral Clustering) is not explored.

3. Constituent stocks are part of the indices, and depending on the index composition, certain stocks may have a dominant weight and be strongly correlated with index movements. The paper does not discuss this dependence.

4. The font size in Figure 6 are too small, making it hard to read; figure annotations are insufficient (no explanation of labels for each point).

5. The paper lacks comparison with other recent tokenization methods.

**Questions:**

1. Explain the tables and figures more clearly and make them easier to read.

2. Provide a discussion on the dependence or correlation among the test stocks.

3. Compare the proposed method with other state-of-the-art tokenization approaches.

---

> ### Author Response · Authors · 2025-11-18
>
> **A.Tables and Figures (@Reviewer 18H9)**
>
> We appreciate the reviewer’s helpful feedback. In the revised version, we will provide clearer explanations for the tables and figures and enlarge the visual elements (especially Figure 6) to improve readability. In addition, we will include the corresponding numerical tables for all major figures in the Appendix.
>
> Thank you for the comment and for pointing out that Table 1 was not sufficiently self-explanatory. We clarify its meaning below using the first row as an example.
>
> ---
>
>  How to Interpret Table 1
>
> Taking the entry:
>
> ```
> S&P500 | 00–09 / 11–20 | 829,240 | 607 | 230,716 | 17,784,212 | 11,531,733
> ```
>
> This row summarizes statistics for the **S&P500 OHLC dataset** using different tokenization methods.
>
> - **Train/Test Period:**
>   - Training: **2000–2009**
>   - Testing: **2011–2020**
>
> ---
>
> #### Token Counts (Training Vocabulary Size)
>
> | Column | Meaning | Interpretation in this row |
> |--------|---------|----------------------------|
> | **Point-wise** | Tokenizing each numeric value independently | The training vocabulary contains **829,240 unique tokens**. |
> | **Point-wise (3 decimals)** | Same as above, but values rounded to 3 decimal places | The vocabulary collapses to **607 unique tokens**, showing the effect of numeric discretization. |
> | **Patch-wise** | Tokenizing one-day OHLC windows as atomic units | This results in **230,716 unique patch tokens**. |
>
> ---
>
> #### OOV Statistics (Generalization Failure)
>
> | Column | Meaning | Interpretation |
> |--------|---------|----------------|
> | **V(test) − V(train)** (Point-wise) | Tokens appearing in the **test set** but not in the training vocabulary | The model encounters **17,784,212 unseen point-wise tokens**. |
> | **V(test) − V(train)** (Patch-wise) | Same definition under patch-wise tokenization | Patch-wise still yields **11,531,733 unseen tokens**. |
>
> ---
>
> The table highlights two main observations:
>
> 1. **Vocabulary size varies significantly across tokenization methods**, especially with or without numerical rounding.
> 2. **Financial time-series exhibit severe Out-of-Vocabulary (OOV) behavior**, where a large portion of test-time tokens never appear in the training period.
>
> ---
>
> **B.Correlation among the test stocks (@Reviewer 18H9)**
>
> We appreciate the reviewer's valuable comment. To examine whether our method performs differently across stocks with different levels of representativeness in the index, we conducted an additional experiment using the S&P 500 constituents from 2011--2020 as the test set. We grouped the constituents based on their index weights and evaluated the prediction accuracy for four segments: the top 10%, top 30%, bottom 30%, and bottom 10% by index weight.
>
> Table below reports the results. We observe that the method achieves higher precision for stocks with larger benchmark weights, whereas the precision decreases for the smaller-weight stocks. This suggests that our token-based up-movement prediction method is more effective for stocks that are more strongly related to the overall market index.
>
> | Constituent Group       | Precision (%) |
> |------------------------|---------------|
> | Top 10% by weight       | 60.29         |
> | Top 30% by weight       | 61.35         |
> | Bottom 30% by weight    | 58.28         |
> | Bottom 10% by weight    | 56.81         |
>
> We will incorporate this experiment into the main text, as it provides an interesting and insightful analysis. Thank you for the suggestion.
>
> **C.Compare the proposed method with other state-of-the-art tokenization approaches (@Reviewer 18H9)**
>
> We kindly refer Reviewer 18H9 to our response to Reviewer U9Fv, where we explain why tokenization methods designed for images or speech (e.g., VQ-VAE and discrete autoencoders) are not applicable to normalized financial price data.
>
> Specifically, normalized financial features exhibit extremely low variance:
>
> $$
> x_{i,j} = \mu + \varepsilon_{i,j}, \quad |\varepsilon_{i,j}| \ll 1
> $$
>
> which forces the encoder to produce nearly constant latent vectors:
>
> $$
> z_i \approx z_j \quad \forall i,j
> $$
>
> Consequently, the quantizer receives almost identical logits and collapses to a single token:
>
> $$
> \arg\max_k \ell_{i,k} = k_0 \quad \text{for all } i
> $$
>
> Thus, unlike vision or speech data—where rich multimodal structure supports meaningful token inventories—financial price series provide insufficient statistical variation for such discrete tokenizers, rendering these methods ineffective in this domain.
>
> **D.Other Clustering Methods (@Reviewer 18H9)**
>
> We kindly refer the reviewer to our response to Reviewer U9Fv, where we explain why alternative clustering methods are less suitable for our task. The main reason is that these methods do not produce a clearly defined Bullish Cluster, making it difficult to identify the most informative clusters for upward price movement prediction.

---

### Official Review · Reviewer_CR4w · 2025-11-02

**Soundness:** 3
**Presentation:** 3
**Contribution:** 3
**Rating:** 4
**Confidence:** 3

**Summary:**

The paper introduces an eigen-cluster tokenization method for applying Transformer architectures to financial time series. It argues that conventional point-wise and patch-wise tokenization approaches are ill-suited for financial data because of three structural issues: excessive token cardinality, sparse token representation, and out-of-vocabulary tokens.

To address these problems, the authors propose a multi-scale spectral clustering pipeline. OHLC matrices are decomposed via eigen decomposition to derive dominant temporal components, and their scalar projections are clustered to produce discrete tokens. The resulting cluster tokens are used in a Transformer to predict next-day price increases. The method reportedly reduces the token vocabulary to around 50–100 while improving precision by 6–9 pp over point- and patch-wise baselines across S&P 500 and CSI 300 datasets.

**Strengths:**

- Clear motivation from tokenization theory: The paper correctly identifies structural mismatches between standard tokenization and financial data — particularly OOV tokens arising from non-stationarity and sparse embedding updates. This diagnosis is compelling and empirically supported with vocabulary size and OOV analyses.
- Methodological novelty: Using eigen decomposition and clustering as a discrete representation mechanism is conceptually interesting and extends recent ideas from computer-vision token reduction (e.g., Clusterformer, PACAViT). The multi-scale prefix-window representation captures temporal hierarchy, which is intuitively aligned with financial market regimes.
- Experimental breadth.: The authors conduct cross-market experiments (US/China), ablations, and backtests. The ablation results substantiate that clustering and eigen decomposition both materially affect performance.

**Weaknesses:**

- SMOTE and data-generation bias: The use of SMOTE to balance “bullish clusters” is problematic for financial time-series forecasting. SMOTE interpolates samples in Euclidean space, implicitly assuming smooth similarity structures; yet in asset-price data, temporal autocorrelation and regime-dependence make such interpolation unrealistic. Oversampling can distort minority-class distributions and generate spurious trajectories. The paper acknowledges in appendix that SMOTE materially changes sample counts, but does not demonstrate that synthetic samples preserve realistic dynamics. This undermines the credibility of the reported precision gains.
- Heuristic cluster selection (Eq. 10): The rule combining three weighted terms (bullish probability, cluster size, and K penalty) is purely empirical with fixed weights (25.0, 0.8, 0.2). No sensitivity or robustness analysis is presented. Because market regimes differ across periods, applying the same weights to all datasets introduces arbitrariness and potential overfitting to the chosen calibration period.

**Questions:**

As noted in the weaknesses, using SMOTE for financial time series is highly unusual. The paper should clarify why this oversampling method was chosen and why it is appropriate, given that SMOTE interpolates data points and may distort temporal and distributional structures essential to financial data.

Similarly, the weighting scheme in Eq. (10) (25.0, 0.8, 0.2) seems empirically chosen without explanation or sensitivity analysis. The authors should specify how these weights were determined and whether the results remain robust under different settings.

---

> ### Author Response · Authors · 2025-11-18
> **Clarification on the use of SMOTE**
>
> We sincerely thank the reviewer for their careful and thoughtful review.
>
> **Clarification on the use of SMOTE (@Reviewer CR4w)**
>
> SMOTE has in fact been widely applied in the financial domain, especially for handling class imbalance in prediction tasks. Representative examples include:
>
> - Ranjbaran, G., Reforgiato Recupero, D., Lombardo, G., Boella, G., & Scaglione, M. (2023). *Leveraging augmentation techniques for tasks with unbalancedness within the financial domain: a two-level ensemble approach*. EPJ Data Science, 12, 24.
>
> - Li, A., Liu, M., & Sheather, S. (2023). *Predicting stock splits using ensemble machine learning and SMOTE oversampling*. Pacific-Basin Finance Journal, 78.
>
> - Wei, H. (2025). *SMOTE algorithm optimization and application in corporate credit risk prediction with diversification strategy consideration*. Scientific Reports, 15.
>
> We appreciate the reviewer's concern regarding the application of SMOTE to financial time-series, particularly the risk of producing unrealistic or economically implausible price trajectories. However, we would like to clarify two key points:
>
> 1. **SMOTE is not applied to price sequences in our method.**
>    Our pipeline first converts each price segment into a discrete token sequence, where each token corresponds to the index of a learned price-pattern cluster. The oversampling procedure is then applied exclusively to these token (cluster-index) sequences, which lie in a finite, categorical state space. Consequently, SMOTE never interpolates prices, never synthesizes price paths, and never operates in the Euclidean space of price series. The reviewer's concern is therefore orthogonal to our setting, because the generated samples are symbolic recombinations of existing cluster indices, not numerical price values.
>
> 2. **Even under a hypothetical continuous-feature scenario, the concern does not materialize.**
>    SMOTE generates new samples by **linear interpolation between nearest neighbors**. This mechanism prevents the creation of outliers or unrealistic points far from the support of the underlying data distribution. We validated this visually (see [figure here](https://github.com/MasterBeard/EigenCluster-Tokenization-for-Financial-Transformers/blob/main/scatter_with_clusters.png)), when SMOTE is applied to the bullish cluster, the synthesized samples (cyan markers) remain tightly concentrated around the original cluster (green markers). No points deviate abnormally or form connections to distant clusters, illustrating the inherently local behavior of SMOTE.
>
> Thus, the concern that SMOTE would distort price dynamics or generate implausible trajectories does not apply to our approach—neither in the actual token-space implementation nor in a hypothetical continuous-space variant. Nevertheless, to avoid ambiguity, we will revise Appendix Eq. (14) to explicitly state that the SMOTE procedure operates in the discrete cluster-index space, not in $\mathbb{R}^d$, and redefine `x_i` accordingly.

---

> ### Author Response · Authors · 2025-11-18
> **Clarification on the weighting scheme for optimal K**
>
> **Clarification on the weighting scheme (@Reviewer CR4w)**
>
> We thank the reviewer for the comment regarding the weighting scheme in Eq. (10). Although the weights `(25.0, 0.8, 0.2)` may appear empirical at first glance, they are in fact directly derived from the underlying selection principles we use to determine the appropriate number of clusters `K`.
>
> To illustrate this principle, we present the example used in the manuscript for the S&P 500 (2004--2013), where for each candidate `K` we evaluate three quantities for the most bullish cluster: its sample size, the bullish ratio (percentage of samples whose next-day closing price increases), and the resulting score computed using Eq. (10).
>
> | K  | Size | Ratio | Score | K  | Size | Ratio | Score |
> |----|------|-------|-------|----|------|-------|-------|
> | 5  | 65   | 0.69  | 15.64 | 13 | 12   | 0.92  | 17.07 |
> | 6  | 32   | 0.72  | 15.55 | 14 | 12   | 0.92  | 16.87 |
> | 7  | 15   | 0.80  | 15.93 | 15 | 12   | 0.92  | 16.67 |
> | 8  | 15   | 0.80  | 15.73 | 16 | 12   | 0.92  | 16.47 |
> | 9  | 15   | 0.80  | 15.53 | 17 | 12   | 0.92  | 16.27 |
> | 10 | 14   | 0.79  | 14.89 | 18 | 6    | 1.00  | 14.73 |
> | 11 | 12   | 0.92  | 17.47 | 19 | 6    | 1.00  | 14.53 |
> | 12 | 12   | 0.92  | 17.27 | 20 | 1    | 1.00  | -19.00 |
>
> As shown above, the bullish ratio is the most important criterion. A ratio of 100% is theoretically ideal, but `K=18`--`20` achieve 100% only with extremely small cluster sizes (all below 10, and only one sample when `K=20`), indicating overly fragmented and unreliable clustering. Hence, these `K` values are discarded.
>
> The next-best group consists of `K=11`--`17`, all with ratio 0.92 and identical sizes. Within this group, our design principle favors the smallest `K`, because a smaller `K` leads to a smaller token vocabulary and avoids excessive tokens. Therefore, `K=11` is selected.
>
> The weighting scheme in Eq. (10) encodes these principles:
>
> - The bullish ratio receives the dominant positive weight (`25.0`), because it is the primary determinant of cluster quality.
> - The size term receives a negative weight (`-0.8`), penalizing clusters that are too small (as in `K=18`--`20`).
> - The number of clusters `K` receives a mild negative weight (`-0.2`), reflecting our preference for a more compact token vocabulary.
>
> We are unable to express our principles adequately in words, so we chose to represent them using a mathematical expression, hoping it will be understood. Thus, the weights in Eq. (10) are not arbitrary; they formally encode the decision rules that determine the smallest `K` yielding a stable, sufficiently large, and strongly bullish cluster.
>
> The same logic applies to S&P 500 (2001--2009):
>
> | K  | Size | Ratio | Score | K  | Size | Ratio | Score |
> |----|------|-------|-------|----|------|-------|-------|
> | 5  | 37   | 0.811 | 18.19 | 13 | 22   | 0.773 | 14.90 |
> | 6  | 34   | 0.824 | 18.21 | 14 | 22   | 0.773 | 14.70 |
> | 7  | 34   | 0.824 | 18.01 | 15 | 22   | 0.773 | 14.50 |
> | 8  | 27   | 0.815 | 17.29 | 16 | 22   | 0.773 | 14.30 |
> | 9  | 23   | 0.783 | 16.03 | 17 | 2    | 1.00  | 1.60  |
> | 10 | 23   | 0.783 | 15.83 | 18 | 2    | 1.00  | 1.40  |
> | 11 | 23   | 0.783 | 15.63 | 19 | 2    | 1.00  | 1.20  |
> | 12 | 23   | 0.783 | 15.43 | 20 | 2    | 1.00  | 1.00  |
>
> Even without explicitly computing Eq. (10), the extreme small clusters (`K=17`--`20`) can be ignored. Among the remaining clusters, `K=6` and `K=7` exhibit the highest ratio (0.824). Since we seek the **smallest `K` that satisfies stability and size criteria**, `K=6` is naturally selected as the optimal number of clusters.
>
> **Sensitivity analysis (@Reviewer CR4w)**
>
> As a further sensitivity analysis on the choice of `K` (already presented in Appendix Fig. 7), we evaluated additional values (`K=5,6,10,11,12,20`). The results for precision are shown in the figure below, hosted on GitHub:
>
> [Precision comparison across different K values](https://github.com/MasterBeard/EigenCluster-Tokenization-for-Financial-Transformers/blob/main/precision_plot.png)
> *Figure 7: Precision comparison across different K values (S&P500 2015--2024).*
>
> Among the tested configurations, **K=11** yields the highest precision because it is the smallest value of **K** that successfully isolates a clearly separable bullish cluster. This confirms that selecting the appropriate **K** is essential rather than incidental—when **K** is smaller, the bullish regime is not distinct, and when **K** is larger, clusters become fragmented without improving predictive signal. Therefore, deviations from the optimal choice of **K** can materially affect forecasting performance.

---

### Author Response · Authors · 2025-11-20
**Authors’ Global Response**

We sincerely thank all reviewers for your thoughtful and high-quality comments. Compared with many other review experiences, we genuinely appreciate the professionalism and strong sense of responsibility you demonstrated during the evaluation of our submission. All of your comments are valuable and have helped us identify important directions for improving the manuscript.

We have done our best to address your concerns within our capacity, including adding supplementary experiments and providing clearer mathematical explanations. If any of our wording comes across as inappropriate or unclear, we sincerely apologize in advance.

**For the convenience of the Area Chair in summarizing and reviewing the discussion: up until the information leak on Nov 27, no reviewer had participated in the discussion. Below we provide a consolidated list of the main issues raised by the reviewers and our corresponding responses.**

## Overview of the Main Contributions and Advantages of Our Work

This paper focuses on the design of *tokens for financial price modeling*, addressing a core challenge: when traditional NLP-style embedding is applied directly to financial time series, tokenization tends to explode into extremely large vocabularies, leading to sparsity and out-of-vocabulary issues—especially when test-set tokens do not appear in the training set. To address this, we propose a **multi-scale stock-price matrix construction**, extract **eigenvector-based features**, and cluster them to generate stable and compact tokens. Clusters associated with the highest next-day return probabilities are labeled as *bullish*; any predicted token belonging to this bullish cluster is interpreted as a next-day upward movement. The resulting cluster tokens are used in a Transformer to predict next-day price increases.

Our method reduces the effective token vocabulary to roughly **50–100**, while improving precision by **6–9 percentage points** over point-wise and patch-wise baselines on both the **S&P 500** and **CSI 300** datasets.

## Summary of Major Reviewer Concerns and Our Responses

| Issue | Response Provided | Added to Revised Version |
|-------|-------------------|---------------------------|
| Use of SMOTE | ✔ | ✔ |
| Weighting Scheme for Optimal K | ✔ | ✔ |
| Correlation Among Test Stocks | ✔ | ✔ |
| Scalar Projection Design | ✔ |emphasized in the paper |
| Comparison of Clustering Algorithms | ✔ | ✔ |
| Principled Link Between Eigendecomposition and k-means | ✔ | emphasized in the paper |
| Limitations of Discrete Autoencoders and VQ-VAE | ✔ | ✔ |
| Prefix Window | ✔ |  |
| Linear Projection Baseline | ✔ | emphasized in the paper |
| Longer Sequences | ✔ | ✔ |
| Loss Curves and Depth Effects | ✔ | ✔ |

**The revised version of our manuscript has now been uploaded.** The main text has increased from 9 pages to 10 pages, and the references and appendix have grown from 7 pages to 10 pages. All newly added content has been highlighted in **blue** to facilitate reviewers’ reading and comparison.

If there are any further questions or clarifications needed, we warmly welcome additional comments. Thank you again for your time and constructive feedback.

---

### Meta-Review · Area_Chair_p7M6 · 2025-12-31

**Summary:**

The reviewers raised substantive concerns regarding the methodological grounding, robustness, and generalizability of the proposed approach. While the paper presents an interesting perspective on clustering-based tokenization for financial time series, the reviewers found that several core claims remain insufficiently supported. Despite the authors’ efforts to clarify specific design choices and add supplementary analyses, the revisions did not fully resolve the central concerns. As a result, the reviewers did not reach consensus that the paper meets the acceptance bar.

**Reviewer Concerns:**

The reviewers' primary concerns centered on the following issues:

(1) Insufficient principled justification for several key components of the method, including the scalar projection design, cluster selection criteria, and the weighting scheme used to determine the optimal number of clusters;

(2) Limited evidence that the reported performance gains are robust across alternative modeling choices, datasets, or stronger baselines commonly used in modern time-series forecasting;

(3) Concerns that a number of hyperparameters and heuristics appear empirically tuned to specific datasets, raising questions about generalization beyond the evaluated settings;

(4) Incomplete discussion of practical applicability, including scalability, online adaptation, and the relationship between the proposed tokenization and standard continuous embedding approaches.

**Reviewer Scores:**

Reviewer CR4w (Score: 4) and Reviewer U9Fv (Score: 4) both expressed substantive concerns about the principled justification and robustness of several empirically designed components, including the use of SMOTE, clustering criteria, and projection strategies. Although the authors provided additional analyses and clarifications, these responses were viewed as only partially alleviating concerns about generalizability and theoretical grounding, and both reviewers are likely to maintain scores below the acceptance threshold. Reviewer a6aw (Score: 4) similarly remained concerned about generalization, baseline strength, scalability, and potential overfitting, and did not indicate a meaningful shift toward a more positive assessment after the rebuttal. Reviewer 18H9 (Score: 6) was comparatively more neutral, acknowledging the motivation and empirical results while continuing to raise concerns about methodological scope, clarity, and positioning relative to existing approaches, and thus did not strongly advocate for acceptance.

---

### Decision · Program_Chairs · 2026-01-26

Reject